# Developmentally regulated H2Av buffering via dynamic sequestration to lipid droplets in *Drosophila* embryos

Matthew Richard Johnson, Roxan Amanda Stephenson, Sina Ghaemmaghami, Michael Andreas Welte*

Department of Biology, University of Rochester, Rochester, United States

**Abstract** Regulating nuclear histone balance is essential for survival, yet in early *Drosophila melanogaster* embryos many regulatory strategies employed in somatic cells are unavailable. Previous work had suggested that lipid droplets (LDs) buffer nuclear accumulation of the histone variant H2Av. Here, we elucidate the buffering mechanism and demonstrate that it is developmentally controlled. Using live imaging, we find that H2Av continuously exchanges between LDs. Our data suggest that the major driving force for H2Av accumulation in nuclei is H2Av abundance in the cytoplasm and that LD binding slows nuclear import kinetically, by limiting this cytoplasmic pool. Nuclear H2Av accumulation is indeed inversely regulated by overall buffering capacity. Histone exchange between LDs abruptly ceases during the midblastula transition, presumably to allow canonical regulatory mechanisms to take over. These findings provide a mechanistic basis for the emerging role of LDs as regulators of protein homeostasis and demonstrate that LDs can control developmental progression.

DOI: https://doi.org/10.7554/eLife.36021.001

## Introduction

Protein homeostasis is essential for cell survival and function, and it is controlled via a plethora of synergizing mechanisms, including protein synthesis, folding, transport, and degradation. Recently, it has become clear that the life cycle of some proteins is substantially modified by lipid droplets (LDs) (*Welte, 2007*; *Welte and Gould, 2017*), cytoplasmic fat storage organelles with critical roles in energy storage (*Hashemi and Goodman, 2015*; *Kühnlein, 2012*; *Pol et al., 2014*; *Sztalryd and Brasaemle, 2017*; *Walther et al., 2017*). For example, LDs are implicated in the refolding of certain damaged proteins in yeast and the assembly of viral protein complexes in mammalian cells. In *Drosophila* embryos, LDs store large amounts of specific histones for use later in development; and in mammals, LDs promote the degradation of certain ER proteins. Although the list of proteins controlled by LDs in some manner continues to expand, the underlying molecular mechanisms remain poorly understood.

Here, we take advantage of the possibly best characterized example of protein handling via LDs, namely their role in histone metabolism in *Drosophila* embryos (*Cermelli et al., 2006*; *Li et al., 2014*, *Li et al., 2012*). Maintaining proper histone stoichiometry within nuclei is critical for cell viability: imbalanced histone deposition onto chromatin is deleterious, resulting in mitotic errors, altered gene expression, and ultimately cell death (*Au et al., 2008*; *Han et al., 1987*; *Kim et al., 1988*; *Meeks-Wagner and Hartwell, 1986*). Control of proper histone levels is also crucial: both overabundance (*Singh et al., 2010*) and dearth (*Celona et al., 2011*) of histones can have profound consequences on the cell. In most cells, histone availability is tightly regulated at multiple levels, including transcriptionally, post-transcriptionally, and post-translationally (*Marzluff et al., 2008*; *Bannister and Kouzarides, 2011*). For example, in mammalian cultured cells, transcription of

***For correspondence:**
michael.welte@rochester.edu

**Competing interests:** The authors declare that no competing interests exist.

canonical histones is highly upregulated during S-phase, when replication occurs, and afterwards throttling of transcription and mRNA turnover bring messages back to very low levels. And in yeast, excess histone proteins are degraded via the ubiquitin-proteasome system (*Singh et al., 2009*).

For early *Drosophila* embryos, regulation of histones is particularly challenging. On the one hand, these embryos develop incredibly rapidly: during the first ~90 min after fertilization, nuclei divide near simultaneously every ~8 min, deep within the embryo (cleavage stages); a subset of these nuclei migrates to the surface and divides four more times over the next hour (syncytial blastoderm stages). All these divisions occur near simultaneously and in the absence of cytokinesis. As a result, histone demand goes up exponentially during this period, in a fluctuating manner, as S-phases and mitoses alternate on minute time scales (the entire interphase is dedicated to DNA replication during these cycles). On the other hand, zygotic transcription is minimal during these stages, being upregulated only in late syncytial blastoderm as part of the midblastula transition (MBT). Thus, early embryos cannot control histone expression transcriptionally. Instead, they draw on abundant histone proteins and mRNAs, produced by the mother during oogenesis. Newly fertilized embryos already contain sufficient histone proteins for thousands of nuclei, and they increase histone levels further by translation of the maternal mRNAs. Thus, histone proteins are consistently overabundant. How is chromatin assembly regulated under those conditions?

Previous work revealed that for three histones, the canonical histones H2A and H2B as well as H2Av, the single H2A variant in *Drosophila*, LDs play an important role in regulating histone availability (*Li et al., 2012*). These histones are bound to LDs via the protein anchor Jabba, and this binding allows embryos to accumulate high levels of maternally provided histones, possibly by protecting them from degradation. This histone storage function allows embryos to develop successfully even when new histone biosynthesis is impaired; without such storage, embryos die under these conditions, in cleavage or syncytial blastoderm stages (*Li et al., 2012*).

In addition to this storage of maternal histones, histone recruitment to LDs also controls nuclear histone accumulation during the syncytial blastoderm. In *Jabba* mutants, embryos accumulate excessive H2Av in nuclei (*Li et al., 2014*). Interestingly, such over-accumulation does not occur with the other LD-associated histones, H2A and H2B, possibly because regulation of canonical histones and histone variants differs dramatically (reviewed in *Baldi and Becker, 2013*). This finding led us to propose that LDs act as H2Av buffers, transiently sequestering H2Av produced in excess to prevent over-accumulation in nuclei. The mechanism of how this buffering is achieved remains unknown. Given how exquisitely many aspects of the histone life cycle are controlled, we hypothesized that the embryo somehow monitors production and usage of H2Av and adjusts release of H2Av from LDs according to the balance at any given moment. To gain insight into the regulation of this process, we set out to monitor transfer of H2Av from LDs to nuclei by live imaging and to quantify H2Av intracellular dynamics.

We find that during cleavage and syncytial blastoderm stages, H2Av is dynamically associated with LDs, constantly exchanging between droplets, with the vast majority of H2Av associated with LDs at any one moment. We propose that this set-up allows the embryo to maintain free H2Av at relatively low concentrations, despite an enormous H2Av pool overall. This small amount of free H2Av limits the availability for nuclear import and subsequent deposition onto chromatin. Thus, buffering is achieved by a passive mechanism that paces incorporation of H2Av into chromatin rather than release of histones from LDs via active feedback. Here, we develop and test a quantitative model for buffering, provide evidence that buffering by LDs is the main regulator of H2Av incorporation during early development, and find that buffering is developmentally regulated, undergoing a dramatic shift at the MBT. Taken together, we elucidate a novel mechanism of histone regulation through sequestration to LDs, providing a paradigm for LDs as key players in protein homeostasis.

## Results

### H2Av can rapidly translocate from LDs to nearby nuclei

Previous work demonstrated a role for LDs as histone stores: when new histone biosynthesis is compromised, histones on LDs are necessary for proper embryonic development (*Li et al., 2012*), presumably to package chromatin because such embryos die with phenotypes reminiscent of severe lack of histones. A critical aspect of this model is that histones can translocate from LDs to nuclei.

Although such transfer has been observed in transplantation experiments (*Cermelli et al., 2006*; *Li et al., 2014*), there is only indirect evidence that it occurs under endogenous conditions (*Li et al., 2012*).

To directly test for such transfer, we employed flies carrying a genomic transgene expressing the photoactivatable fusion protein H2Av-paGFP (*Post et al., 2005*). Embryos from such mothers are viable and exhibit normal chromosome dynamics (*Post et al., 2005*); a similarly engineered H2Av-eGFP fusion protein rescues an *H2Av* null allele and is a widely used in vivo marker of chromosome behavior (*Clarkson and Saint, 1999*). To follow specifically the fate of LD-associated H2Av-paGFP, we focused on late syncytial blastoderm embryos, as here presumably demand for histones is the highest. The embryo in *Figure 1* is in nuclear cycle 13 (NC13), that is in between mitosis 12 and mitosis 13, the last syncytial cycle. We photoactivated a region enriched for LDs and monitored fluorescent signal over time. Within minutes, signal disappeared from the photoactivated region and appeared in nuclei, specifically those directly adjacent to the activation region (*Figure 1*). This experiment demonstrates that histone H2Av can rapidly translocate from LDs to nearby nuclei under endogenous conditions.

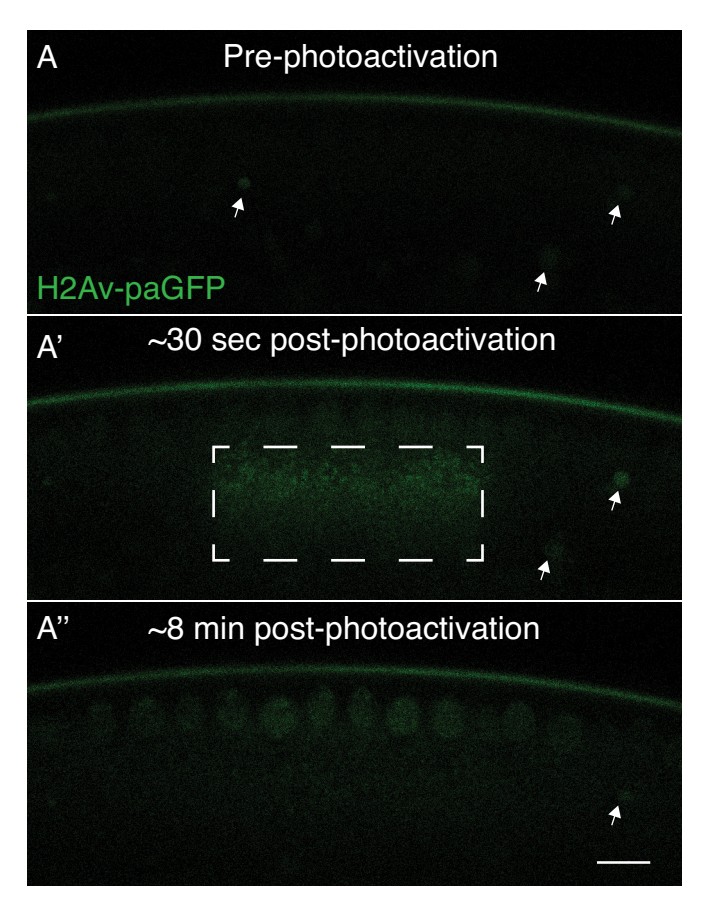

**Figure 1.** H2Av can translocate from lipid droplets (LDs) to nearby nuclei. In NC13 embryos expressing H2Av-paGFP, a region enriched for LDs was photoactivated by exposure to 405 nm light (Dashed box, A'). (**A**) Prior to photoactivation, (**A'**) ~30 s post-photoactivation, (**A''**) ~8 min post-photoactivation. The majority of H2Av-paGFP dissipates from LDs and accumulates preferentially in nuclei immediately adjacent to the photoactivated region. White arrowheads indicate autofluorescent yolk particles within the embryo and are not indicative of H2Av-paGFP signal. Scale bar represents 10 μm.
DOI: https://doi.org/10.7554/eLife.36021.002

## H2Av is rapidly lost from LDs in late syncytial blastoderm embryos

The extent to which H2Av was lost from LDs was unexpected: within just minutes, nearly all signal had dissipated from the region of initial activation. We hypothesized that this dramatic loss reflects the massive demand for histones in the thousands of nuclei of late blastoderm embryos. If this notion is correct, loss of H2Av from nuclei should roughly scale with the number of nuclei present in the syncytial embryo: slow to non-existent in embryos with few nuclei, and occurring increasingly rapidly with each nuclear division, with maximal rates in NC13 and 14 when thousands of nuclei simultaneously undergo S phase (such as in the embryo in *Figure 1*). We therefore set out to compare the rate of loss of H2Av from LDs at different embryo ages.

This analysis is complicated by the fact that the spatial distribution of LDs across the embryo changes dramatically during embryogenesis. We therefore employed fly lines expressing H2Av-Dendra2 from a genomic transgene (*Figure 2A*): this strain is wild type at the endogenous *H2Av* locus, that is, H2Av-Dendra2 is expressed in addition to endogenous, untagged, H2Av. Unless otherwise indicated, all subsequent experiments utilizing H2Av-fusion proteins also employ flies wild type at the endogenous *H2Av* locus.

Before photoswitching, Dendra2 fluoresces green when excited with blue light (similar to GFP); after photoswitching, it fluoresces red when excited with green light (similar to RFP) (*Gurskaya et al., 2006*). This property provides two advantages: First, we can unambiguously determine the location of LDs before activation and thus target activation to a precise area (*Figure 2B,C*), and second, it is possible to track the behavior of both photoswitched (red) and non-photoswitched (green) H2Av populations.

In embryos expressing H2Av-Dendra2, we performed photoswitching in NC13 embryos near the embryonic surface, just below the nuclear layer (*Figure 3A*). Consistent with our previous observations using H2Av-paGFP (*Figure 1*), H2Av-Dendra2 was rapidly lost from the photoswitched region (*Figure 3B,C*, *Figure 3—figure supplement 1A*). Quantification revealed that over the course of 3 min, approximately 50% of the original signal had dissipated from the region of activation.

## Loss of H2Av from LDs is independent of histone demand

A conclusive assessment of the idea whether H2Av loss from LDs is influenced by histone demand requires a quantitative comparison across development. We therefore measured the rate of loss from the photoswitched regions in cleavage stage embryos, when nuclear number is dramatically less than in NC13. Surprisingly, H2Av-Dendra2 was lost from LDs at similar rates as we observed in NC13 (*Figure 3C*), in stark contrast to our hypothesis. This loss was accompanied by signal appearing in adjacent regions, indicating lateral spread; a similar spread is evident in NC13 (*Figure 3B*, *Figure 3—figure supplement 1A*).

As an independent confirmation of this unexpected result, we monitored the behavior of the population of non-photoswitched H2Av-Dendra2; originally depleted from the photoswitched region, it returned with similar kinetics as the photoswitched population, presumably supplied by the surrounding regions (*Figure 3—figure supplement 1B*). We observed the same behavior in experiments performing fluorescence recovery after photobleaching (FRAP) using H2Av-mRFP (*Figure 4A–C*, middle). Thus, even in early embryos, the spatial distribution of H2Av is highly dynamic.

We next measured the rate of loss from the photoswitched regions throughout early embryogenesis, from unfertilized embryos through early NC14, representing ~2 hr of developmental time. Across all these stages, rates of loss are remarkably similar (*Figure 3C*).

To assess if the spread of H2Av signal was due to the motion of entire LDs, we marked LDs using LSD2-YFP; LSD-2 is a member of the Perilipin-family of LD-associated proteins (*Welte et al., 2005*). In FRAP experiments, signal recovery was minimal; for example, in embryos expressing both H2Av-mRFP and LSD2-YFP, H2Av-mRFP recovery was dramatically higher than LSD2-YFP (*Figure 4A–D*). We also monitored LD motion directly, by recording LSD2-YFP signal over time; in most cases, we detected very little motion (*Video 1*), consistent with previous reports that LD motion is minimal in cleavage stage embryos (*Welte et al., 1998*). In some instances, large-scale movements within the embryo were observed, likely due to cytoplasmic streaming (*von Dassow and Schubiger, 1994*). However, even in these embryos, the bleached region moved largely as a unit and remained devoid of YFP signal, demonstrating that even large-scale movement within the embryo is not sufficient to

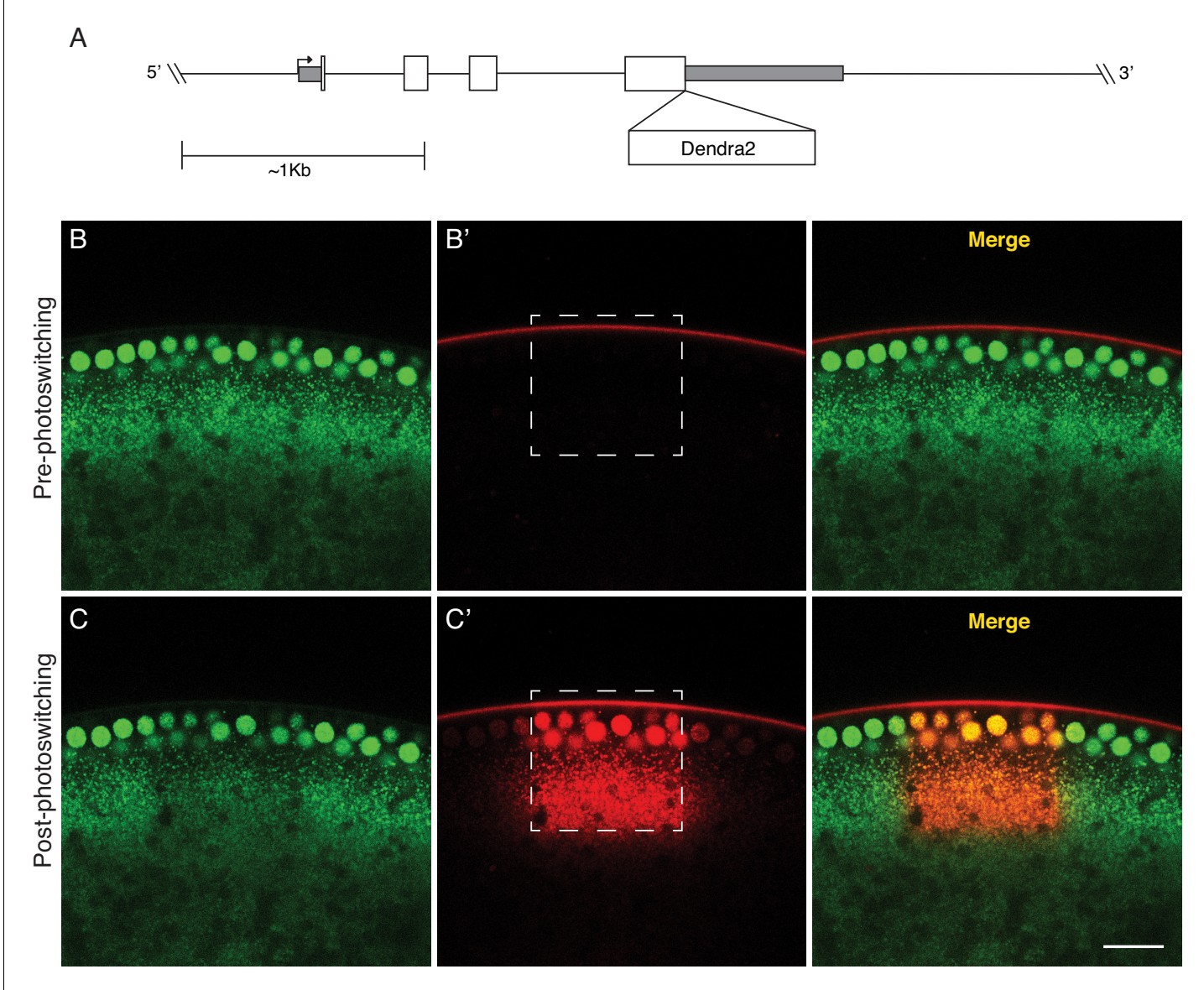

**Figure 2.** A photoswitchable H2Av-Dendra2 for in-vivo H2Av tracking. (**A**) Generation of a transgene containing H2Av-Dendra2 under endogenous regulation. The endogenous *H2Av* genomic region (~4 Kb) was amplified via PCR and the Dendra2 protein coding sequence (Evrogen) was inserted, in frame, at the end of the fourth exon, prior to the stop codon. White boxes represent coding exons. Grey boxes represent UTRs. Scale represents approximately 1 Kb. (**B,C**) Expression of H2Av-Dendra2 in NC13 embryos. Embryos expressing H2Av-Dendra2 prior to (**B**) and after (**C**) photoswitching. Different channels represent non-photoswitched H2Av-Dendra2 (green, [**B,C**]) and photoswitched H2Av-Dendra2 (red, [**B', C'**]). Substantial photoswitching is achieved via exposure of a region of interest (dashed box) to 405 nm light. Scale bar represents 20 µm.
DOI: https://doi.org/10.7554/eLife.36021.003

explain the observed H2Av dynamics (*Figure 4—figure supplement 1*). We conclude that even in embryos with few nuclei, H2Av is rapidly lost from LDs.

Taken together, we find that the dynamics of H2Av association with LDs is similar throughout early embryogenesis. These results are not compatible with the idea that loss of H2Av from LDs is driven by demand for histones due to ongoing replication. We conclude that loss of H2Av from LDs is independent of histone demand.

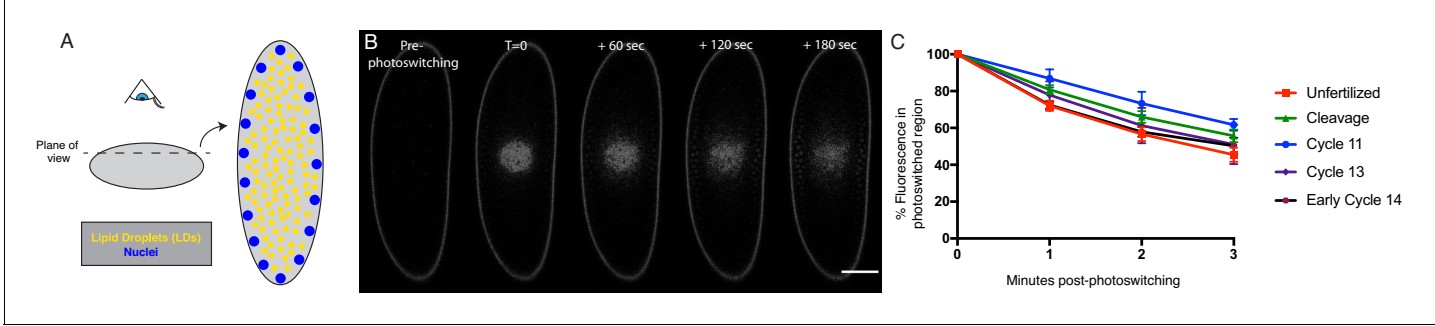

**Figure 3.** H2Av is rapidly lost from LDs throughout early embryogenesis. (**A**) Scheme of experimental design. (Left) Photoswitching of LD-enriched regions is achieved by focusing just below the embryo surface. Dashed line represents plane of view. (Right) When viewing the embryo at this focal plane, nuclei are visible at the periphery (blue) and LDs are present throughout the focal plane (yellow). (**B**) After photoswitching, H2Av-Dendra2 signal is rapidly lost from the photoswitched region, appearing to spread throughout the embryo. Red channel (photoswitched H2Av-Dendra2) is shown. Images of NC13 embryos were taken every 60 s post-photoswitching (See also *Figure 3—figure supplement 1*). Scale bar represents 50 μm. (**C**) Quantitation of fluorescent signal within the photoswitched region shows rapid loss of signal over time. Values represent the percentage of initial fluorescence remaining within the photoswitched region at the indicated time points. Colors represent different ages of embryos. Five embryos per stage were quantified. Error bars represent SD.

DOI: https://doi.org/10.7554/eLife.36021.004

The following figure supplement is available for figure 3:

**Figure supplement 1.** H2Av is dynamically associated with LDs.

DOI: https://doi.org/10.7554/eLife.36021.005

## H2Av exchanges between LDs, likely via a cytoplasmic route

What happens to the H2Av lost from LDs? Previous transplantation experiments had found that exogenously provided H2Av can localize to LDs (*Li et al., 2014*). In both photoswitching and FRAP experiments, signal that has spread to new regions is not uniformly distributed but displays a punctate appearance (e.g. *Figure 3B*, *Figure 4A–C*). When both photoswitched H2Av-Dendra2 (red) and non-photoswitched H2Av-Dendra2 (green) are monitored, colocalization to the same puncta increases with time after photoswitching (*Figure 4E–G*). Since at steady state the vast majority of H2Av signal in early embryos is detected on LDs (*Cermelli et al., 2006*; *Li et al., 2012*), these observations strongly suggest that H2Av lost from LDs are gained by other LDs. Indeed, at higher magnification, we detect colocalization of photoswitched and non-photoswitched H2Av-Dendra2 in the ring-pattern characteristic of LD proteins (*Figure 4G′*). We conclude that H2Av constantly exchanges between LDs.

In early embryos, LDs are broadly distributed and show very little motion (*Video 1*). Thus, it seems unlikely that H2Av is exchanged between LDs via direct contact. We hypothesize that H2Av travels between LDs through the cytoplasm, although have not been able to unambiguously detect H2Av signal in the cytoplasm above background. We employed in vivo centrifugation to separate LDs from other embryonic components, including the cytoplasm (*Cermelli et al., 2006*; *Tran and Welte, 2010*). When pre-cellular embryos are centrifuged, they become stratified, with cellular components sorting out by density; in particular, LDs accumulate at the end of the embryo pointing up, forming a distinct layer. When we induced photoswitching in the LD layer of centrifuged H2Av-Dendra2 embryos, we detected robust activation, demonstrating that photoswitching still works under these conditions (*Figure 4I*). When we performed photoswitching away from the LD layer, a weak, but specific, H2Av-Dendra2 signal was detected, indicating the presence of a potential non-LD associated H2Av pool (*Figure 4I′*). We have not noticed any particular spatial structure to this signal, and it does spread rapidly after activation (data not shown). Although these data do not completely rule out the involvement of other cellular compartments (including a potential small population of LDs of aberrant size or density), we propose that a small, non-LD pool of H2Av is present in the cytoplasm.

## Does H2Av exchange between LDs mediate the buffering role of LDs?

We had previously proposed that LDs serve as H2Av buffers (*Li et al., 2014*) because H2Av is likely produced in excess and when recruitment to LDs is abolished, it over-accumulates in nuclei of

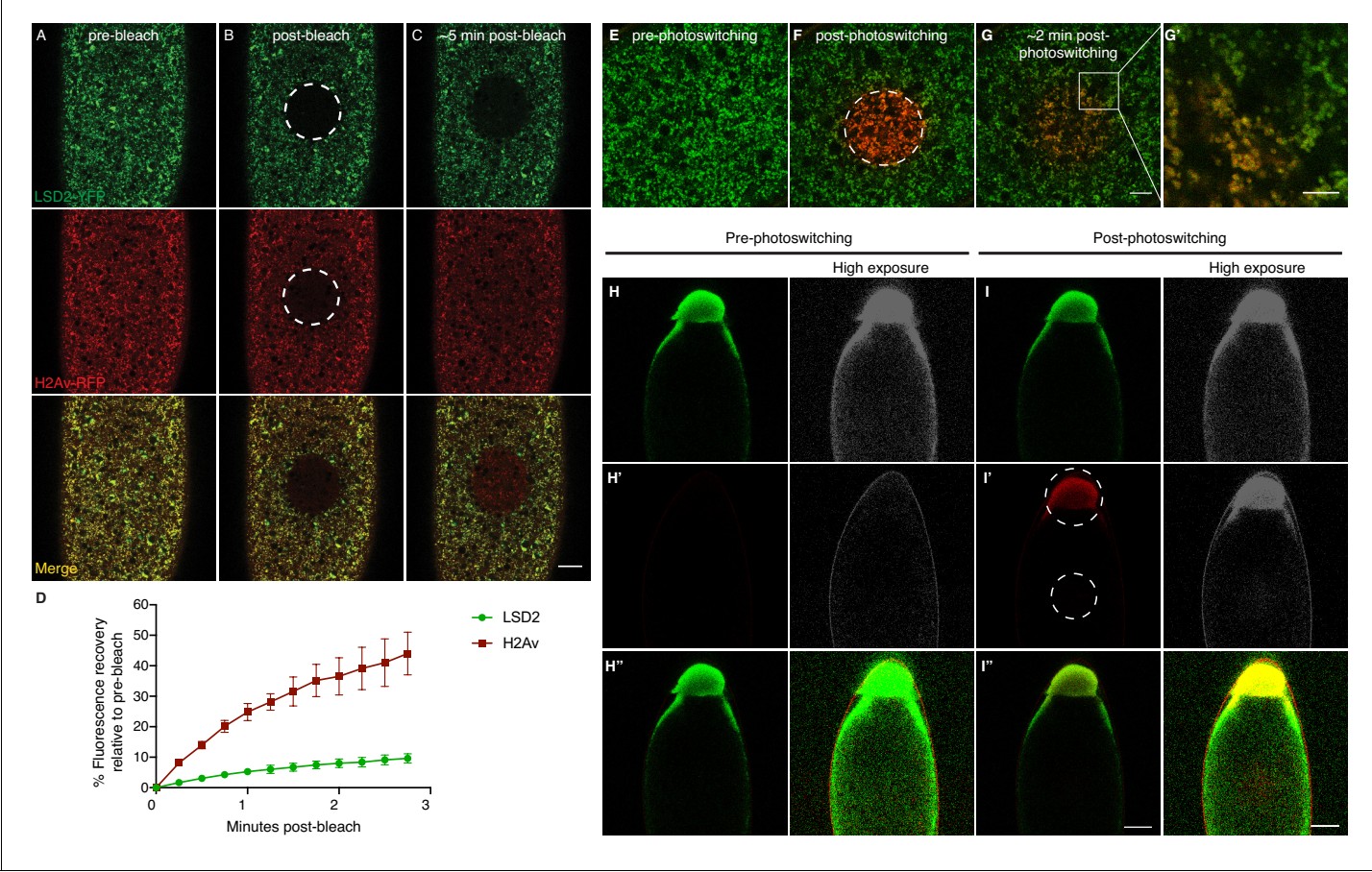

**Figure 4.** H2Av exchanges between LDs via a cytoplasmic route. (**A–C**) Fluorescence Recovery After Photobleaching (FRAP) in cleavage stage embryos expressing LSD2-YFP (green), a LD marker, and H2Av-mRFP. Photobleaching was induced in LD enriched regions near the embryonic surface (dashed circle) and recovery was monitored over time. (**A**) Pre-bleach, (**B**) immediately post-bleach, (**C**) ~5 min post-bleach. Scale bar represents 20 μm. (**D**) Quantitation of FRAP experiments in A-C. N = 3. Error bars represent SD. See also *Figure 4—figure supplement 1*. (**E–G**) Photoswitching was induced in cleavage stage embryos expressing H2Av-Dendra2. Within minutes, colocalization (yellow) of non-photoswitched (green) and photoswitched (red) H2Av-Dendra2 is evident. Scale bar represents 10 μm. (**G'**) Colocalization shows the ring pattern characteristic of LD proteins. Scale bar represents 5 μm. (**H–I**) Photoswitching in centrifuged embryos expressing H2Av-Dendra2. Photoswitching was induced either at the LD layer or in cytoplasmic regions (dashed circles, **I'**). A weak, but specific, cytoplasmic population of H2Av-Dendra2 is evident under high exposure. In centrifuged wild-type embryos, no signal is observed after similar exposure to 405 nm light (not shown). Scale bar represents 10 μm.

DOI: https://doi.org/10.7554/eLife.36021.006

The following figure supplement is available for figure 4:

**Figure supplement 1.** Observed H2Av dynamics do not result from cytoplasmic streaming.

DOI: https://doi.org/10.7554/eLife.36021.007

syncytial blastoderm embryos. The mechanism of buffering was unknown, but our working hypothesis had been that capture by or release from LDs depends on the overall need for H2Av at any particular moment: the embryo somehow monitors production and usage of H2Av and the balance between those two processes controls the H2Av dynamics at the LDs. This model proposes that capture or release are controlled by feedback regulation. For core histones, it has been shown that a plethora of mechanisms synergize to indeed carefully coordinate core histone production and deposition (*Figure 5—figure supplement 1*). For example, if in cultured mammalian cells DNA replication is inhibited during S phase, histone messages are prematurely degraded, suggesting that reduced demand for histones triggers a feedback loop that curbs histone production (*Baumbach et al., 1984*).

In contrast to such a highly controlled process, our discovery of the dynamic and seemingly unregulated nature of H2Av association with LDs suggests a radically different alternative: newly

produced H2Av can reversibly bind to LDs, which reduces the concentration of H2Av free in the cytoplasm and thus the rate of H2Av import into nuclei. In this view, nuclear H2Av levels are controlled kinetically, by slowing how fast H2Av enters nuclei, not by feedback control that adjusts H2Av release from LDs to demand.

Delivery of newly synthesized H2Av to chromatin must involve many molecular players, including histone chaperones in both the cytoplasm and the nucleoplasm, the nuclear import machinery, and other histones (as H2Av forms heterodimers with H2B). In an extreme version of the model, however, none of these steps would be regulated or limiting under the conditions of the early embryo (*Figure 5A*). To determine to what extent such a stripped-down model is able to explain the situation in *Drosophila* embryos, we set out to test key assumptions.

## Export of H2Av is negligible compared to import

One of the implicit assumptions of the model in *Figure 5* is that nuclear H2Av levels are mostly controlled by gain of H2Av from the cytoplasm, with minimal loss of H2Av from nuclei. We therefore compared gain and loss of H2Av using FRAP and photoactivation. When H2Av-GFP was bleached in nuclei during interphase, signal recovered steadily over the next few minutes (*Figure 6A,B*), indicative of constant H2Av-GFP nuclear import. In contrast, when we partially photoswitched single nuclei in embryos expressing H2Av-Dendra2, we did not detect any significant loss of signal over several minutes, although non-photoswitched H2Av-Dendra2 increased in these nuclei (*Figure 6C,D*), consistent with continued delivery of H2Av-Dendra2 from LDs, as in the experiment in *Figure 6A,B*. We conclude that import of H2Av is vastly more prominent than loss of H2Av and thus should be the main determinant of H2Av levels in the nuclei.

We suspect that a major reason why H2Av export is minimal is that once incorporated into chromatin, H2Av is irreversibly trapped. Consistent with this notion, relative H2Av signal on metaphase chromosomes correlates with total nuclear H2Av levels in the preceding interphase, both values increasing from NC11 to NC13 (*Figure 7A,B*). In addition, almost all of the H2Av present in mitotic nuclei of *Jabba* embryos localized to chromosomes (*Figure 7D,E*), even though those H2Av levels are significantly increased compared to wild type (*Li et al., 2014*).

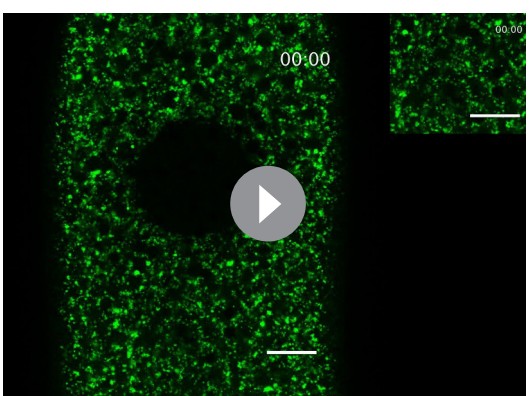

**Video 1.** LDs show little movement and minimal LD-LD contact during cleavage stages. Video shows a composite of two sequences from the same embryo. Images were taken every 15 s. Playback speed is 7 frames/s. Left: video showing a FRAP experiment in embryos expressing the LD marker LSD2-YFP. Scale bar represents 20 μm. Top right video highlights a region of the embryo away from the bleached region. Although little motion is observed in these early stages, LDs move actively along microtubules from NC10 onwards (*Welte, 2015*). Even then, however, motion is perpendicular to the plane of focus shown here and thus would not account for lateral spread or exchange within the plane.

DOI: https://doi.org/10.7554/eLife.36021.008

## Nuclear H2Av levels reflect available H2Av

If none of the steps in *Figure 5—figure supplement 1* are rate-limiting, including nuclear import, then nuclear H2Av accumulation should directly depend on how much H2Av is available in the cytoplasm. For many proteins, a simple way to alter overall protein levels is by varying the copy number of the corresponding gene. *A priori*, it seemed unlikely that this approach would have a profound effect on total histone protein levels: for core histones, elaborate feedback mechanisms keep histone levels constant despite huge variation in gene copy number. For example, both mRNA and protein levels for these histones are indistinguishable in early embryos from females carrying either 24 or ~200 gene copies (*McKay et al., 2015*).

However, when we varied the number of *H2Av* genes (using a null allele or genomic transgenes) from one to four, global H2Av proteins levels – as determined by western analysis – were roughly proportional to gene dosage (*Figure 8A,B*). Our results not only underscore that H2Av expression

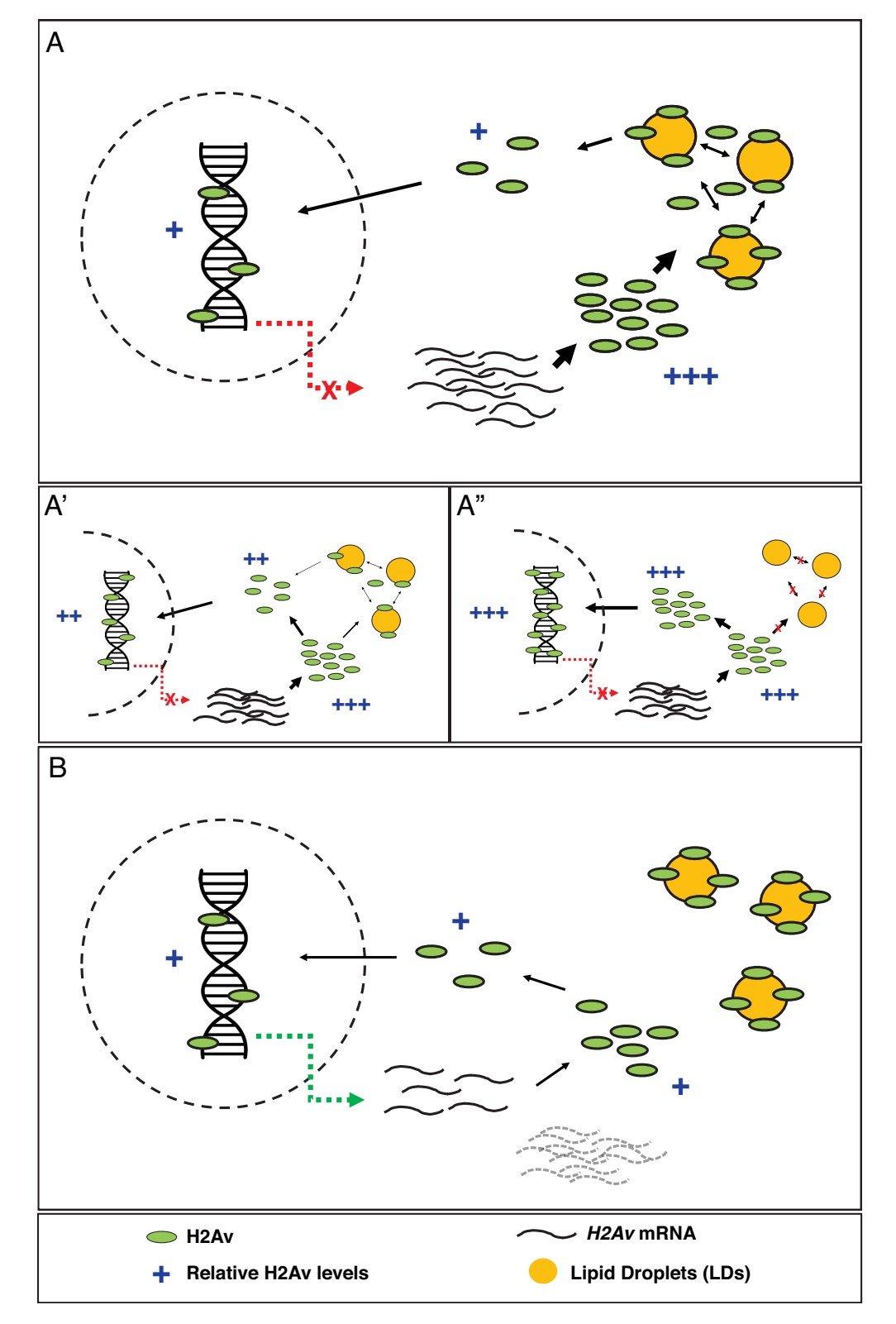

**Figure 5.** Model: LDs constitute the major regulator of H2Av protein levels both free in the cytoplasm and in the nucleus. (A) In early *Drosophila* embryos, LDs are the main H2Av regulator. *H2Av* mRNA levels are stable; these represent maternal mRNAs, and there is no zygotic contribution (red dashed line). H2Av synthesized in excess is buffered by LDs, limiting the amount of H2Av available free in the cytoplasm for subsequent nuclear import and deposition onto chromatin. (A',A") Reduction in buffering capacity reduces the amount of available H2Av, resulting in reduced nuclear import and

*Figure 5 continued on next page*

eLIFE Research article

Cell Biology | Developmental Biology

*Figure 5 continued*

deposition onto chromatin. (**A'**) *1x Jabba* and (**A''**) *0x Jabba*. (**B**) H2Av buffering by LDs is dramatically reduced at the midblastula transition (MBT). Maternal *H2Av* mRNA stores (black dashed lines) are degraded, and zygotic *H2Av* transcription is activated (green dashed line), resulting in overall lower H2Av protein production. LD mediated H2Av buffering ceases, and H2Av is neither lost from nor transiently sequestered to LDs. Width of black arrows indicate H2Av flux at each step of regulation. See also *Figure 5—figure supplement 1*.

DOI: https://doi.org/10.7554/eLife.36021.009

The following figure supplement is available for figure 5:

**Figure supplement 1.** In most cells, histones are regulated at multiple steps.

DOI: https://doi.org/10.7554/eLife.36021.010

is regulated very differently from that of canonical histones, but also suggest that translational and post-translational regulatory mechanisms do not compensate to achieve a specific overall level of H2Av protein in the early embryo.

These findings allowed us to test whether and how H2Av availability affects accumulation of H2Av in the nucleus. In syncytial blastoderm stages, embryos with two copies of *H2Av* (2x *H2Av*) have elevated levels of nuclear H2Av compared to equally staged embryos with a single copy of *H2Av* (1x *H2Av*), and nuclear levels are further increased in embryos with four copies (4x *H2Av*) (*Figure 8C,D*). These findings support the notion that nuclear H2Av accumulation reflects overall H2Av availability. And consistent with this idea, in 2x *H2Av* embryos where H2Av cannot bind to LDs (*i.e.* mutant for *Jabba*) and thus cytoplasmic H2Av levels are presumably increased, nuclear H2Av levels are substantially elevated compared to 2x *H2Av* embryos (*Li et al., 2014*).

## Devising a kinetic model for LD-mediated H2Av buffering

Although it is intuitive that for the model in *Figure 5* more available H2Av should result in higher accumulation of H2Av in nuclei, it is less obvious by how much nuclear H2Av will increase. To understand this relationship, we developed a formal kinetic model for histone buffering (*Figure 9*, *Figure 9—figure supplement 1*, Appendix 1), based on principles of equilibrium thermodynamics; this model assumes that on the time scales relevant for our observations the system is in an apparent equilibrium. We considered H2Av as existing in three pools: free in the cytoplasm, LD-bound, and DNA-bound. Free H2Av constantly exchanges with DNA-bound H2Av, with on-rate $k^2_{on}$ and off-rate $k^2_{off}$ (*Figure 9A* top, right). This system is governed by the law of mass action, and at any one moment the net flux of H2Av into the nucleus is dependent on the concentrations of free cytoplasmic H2Av. The level of free cytoplasmic H2Av, in turn, is dependent on the concentration of LDs (Appendix 1). If no LDs are present, all H2Av is free in the cytoplasm and the rate of import into the nucleus is at its maximum (*Figure 9B*). As the level of LDs increase, H2Av becomes rapidly partitioned into free and LD-bound pools upon synthesis, lowering the rate of import into the nucleus (*Figure 9C*).

The details of the model, and the exact mathematical relationship between the rate of nuclear import and the concentration of LDs, are outlined in Appendix 1. Briefly, we devised three rate equations describing changes in concentrations of free, LD-bound, and DNA-bound H2Av. We then solved these equations by making a number of simplifying assumptions (Appendix 1). The three most critical ones are graphically depicted in *Figure 9A* (bottom): First, we assumed that H2Av bound to DNA is significantly lower in energy than H2Av bound to LDs. This assumption is based on the observation that there is net flux from LDs to nuclei (*Figure 1*), but not in the reverse direction (*Figure 6D*). Second, we assumed that H2Av bound to LDs is significantly lower in energy than H2Av free in the cytoplasm. In support of this notion, we find the concentration of H2Av bound to LDs is dramatically higher than in the cytoplasm (*Figure 4H*). Third, the on-rate of H2Av binding to LDs ($k^1_{on}$) is faster than its incorporation into nuclear DNA ($k^2_{on}$). This assumption is based on the observation that H2Av lost from LDs (and thus having passed through the cytoplasmic pool) efficiently accumulates on LDs again (*Figure 4G'*). Thus, the model assumes that there is a rapid pre-equilibration of H2Av between the cytoplasm and LDs (*Figure 9A*, bottom, left). At shorter time-points, this reduces H2Av incorporation into nuclei. However, over longer periods of time, gradual accumulation of nuclear H2Av occurs, as its relative energy is lower than that of H2Av bound to LDs (*Figure 9A*, bottom, right).

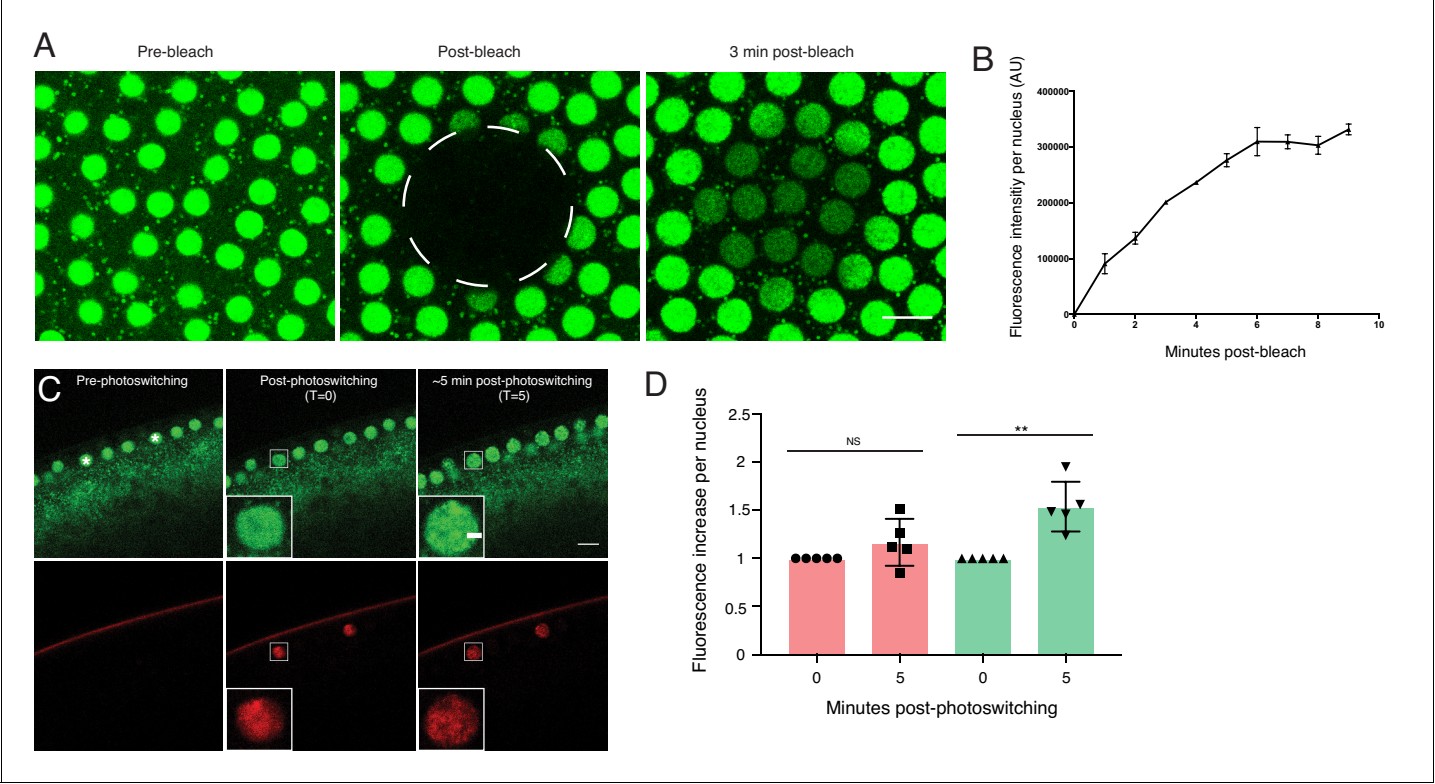

**Figure 6.** Nuclear H2Av levels are driven by nuclear import. (**A,B**) A region encompassing several nuclei was photobleached in NC13 embryos expressing H2Av-GFP, and nuclear signal was monitored over time. (**A**) Representative images of a FRAP experiment in NC13 embryos. White dashed circle indicates area of bleaching. Scale bar represents 10 µm. (**B**) Quantitation of fluorescence intensity per nucleus after bleaching. Five nuclei were averaged per embryo. N = 3 embryos. (**C**) Photoswitching was induced in single nuclei (white asterisks) in NC13 embryos. Scale bars represent 10 and 2 µm (inlay). (**D**) Quantitation of changes in fluorescent signal within individual nuclei ~5-min post-photoswitching. Two nuclei were averaged per embryo, N = 5 embryos. For T = 0 vs. T = 5 in the red channel, p=0.1639. For T = 0 vs. T = 5 in the green channel, p=0.0017. All error bars represent SD. All p values were calculated using an unpaired student t-test.

DOI: https://doi.org/10.7554/eLife.36021.011

With these assumptions and boundary conditions, the non-nuclear pool of H2Av (i.e. H2Av on LDs and free in the cytoplasm) decays exponentially with time according to the following expression (Appendix 1):

$$[H_{tot}] \cdot e^{\frac{K_{HL}}{K_{HL}+[L]} \cdot [D] \cdot k_{on}^2 \cdot t}$$

Here, $K_{HL}$ is the equilibrium constant for H2Av binding to LDs (i.e. $k^1_{off}/k^1_{on}$), [L] the concentration of H2Av binding sites on LDs, and [D] the concentration of H2Av binding sites on DNA. DNA-bound H2Av (1) is the remainder of the H2Av pool and thus its accumulation as a function of time is as follows (Appendix 1):

$$HD(t) = [H_{tot}] \cdot \left(1 - e^{\frac{K_{HL}}{K_{HL}+[L]} \cdot [D] \cdot k_{on}^2 \cdot t}\right) \qquad (1)$$

This curve starts at 0 for t = 0, reflecting the chosen boundary conditions, and asymptotically approaches [H_{tot}], reflecting that at infinite time all H2Av will be imported into the nuclei.

In particular, this equation predicts that an increase in total H2Av abundance ([H_{tot}]) causes a proportional increase in nuclear H2Av (*Figure 9—figure supplement 1A*). Indeed, we found that nuclear H2Av levels increase linearly with elevated H2Av abundance (*Figure 8D*).

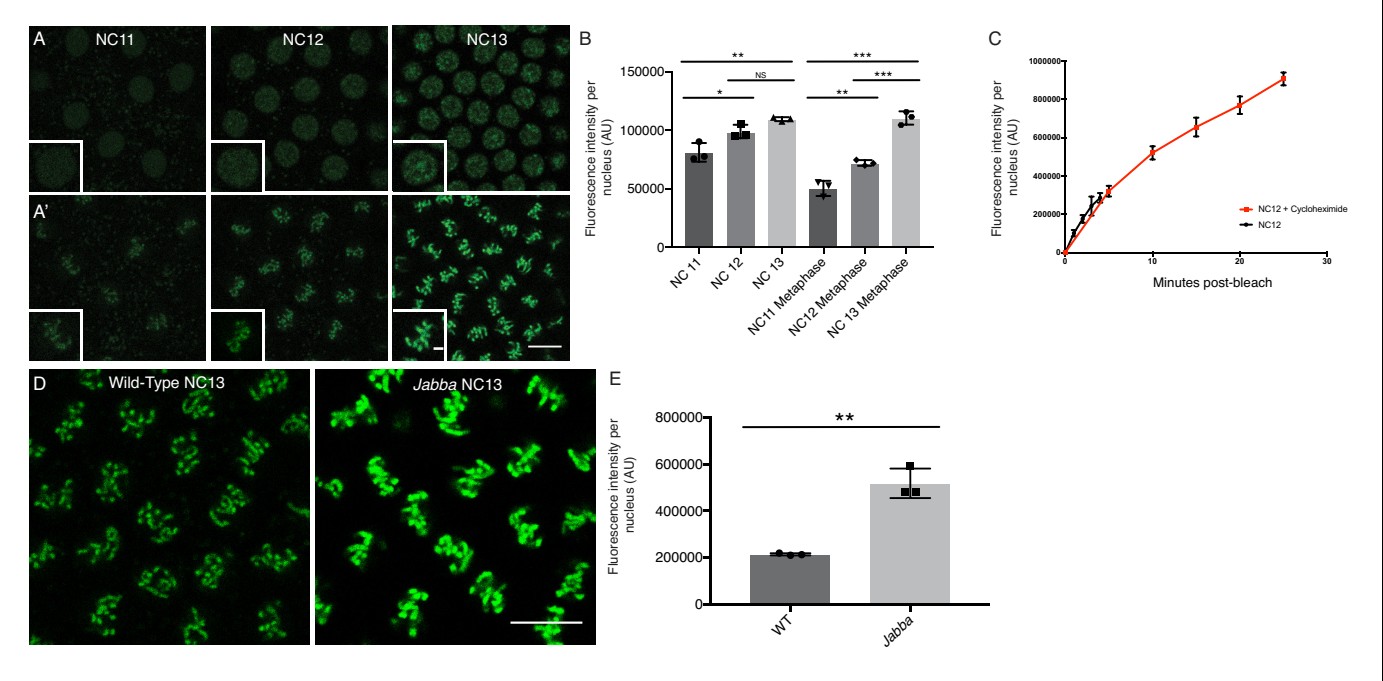

**Figure 7.** Nuclear H2Av is chromosome associated and scales with interphase length. (**A**) NC11-NC13 embryos expressing H2Av-GFP in interphase (**A**) or metaphase (**A'**). Scale bars represent 10 and 2 μm (inlay). (**B**) Quantitation of total nuclear H2Av-GFP fluorescent signal per nucleus in interphase and metaphase of NC11-NC13 embryos. Five nuclei were averaged per embryo, N = 3 embryos. NC11 vs NC12, p=0.0254; NC11 vs. NC13, p=0.0048; NC12 vs NC13, p=0.1214, NC11 metaphase vs. NC12 metaphase, p=0.0060; NC11 metaphase vs. NC13 metaphase, p=0.0001; NC12 metaphase vs. NC13 metaphase, p=0.0007. All p values were calculated using a two-way ANOVA followed by Tukey's test. (**C**) Quantitation of FRAP experiments performed as described in *Figure 6A* on NC12 embryos injected with cycloheximide (red) or not (black). Nuclear levels continue to rise throughout a prolonged interphase. Five nuclei were averaged per embryo, N = 3 embryos. (**D**) Relative H2Av-GFP signal on metaphase chromosomes in NC13 wild-type and *Jabba* embryos. Scale bar represents 10 μm. (**E**) Quantitation of total signal within an equally sized ROI encompassing individual metaphase chromosomes. Five nuclei were averaged per embryo, N = 3 embryos. WT vs. *Jabba*, p=0.0011. All p values were calculated using an unpaired student t-test. All error bars represent SD.

DOI: https://doi.org/10.7554/eLife.36021.012

## Nuclear H2Av levels depend on LD buffering capacity

A major prediction of our quantitative modeling is that increasing the amount of H2Av binding sites on LDs ([L]) will decrease the rate of formation of DNA-bound H2Av (*Figure 9—figure supplement 1B*). To modulate this LD buffering capacity (i.e. modulate H2Av-binding sites on LDs), we reduced Jabba protein levels, taking advantage of *Jabba* null alleles. In embryos expressing 0, 1, or 2 copies of *Jabba*, we observed a corresponding increase in Jabba protein per embryo (*Figure 10A,B*). Previous experiments had demonstrated that elevated *Jabba* dosage is sufficient to increase H2Av levels on LDs (*Li et al., 2012*): centrifuged embryos laid by wild-type mothers had substantially higher H2Av-GFP levels localized to the LD layer than embryos laid by mothers expressing only 1 copy of *Jabba*. In embryos expressing H2Av-GFP, we then compared nuclear H2Av levels in NC13 embryos of varying *Jabba* dosage (at the same time during the cell cycle). Increased *Jabba* dosage is accompanied by reduced nuclear accumulation of H2Av-GFP (*Figure 10C,D*).

To directly test whether altered nuclear H2Av levels with varying *Jabba* dosage resulted from changes in nuclear import rates, we performed FRAP experiments in NC13 embryos of varying *Jabba* dosage, including over-expressing Jabba using a single copy of a genomic *Jabba* transgene (3x *Jabba*). As predicted, increases in *Jabba* dosage from 1x to 2x and 3x slowed H2Av nuclear import (*Figure 10E*). Furthermore, when import rate constants are plotted as a function of relative Jabba expression, the fitted curve matched the relationship predicted by our kinetic model (*Figure 10F*). We conclude that overall buffering capacity inversely affects the rate of H2Av nuclear import.

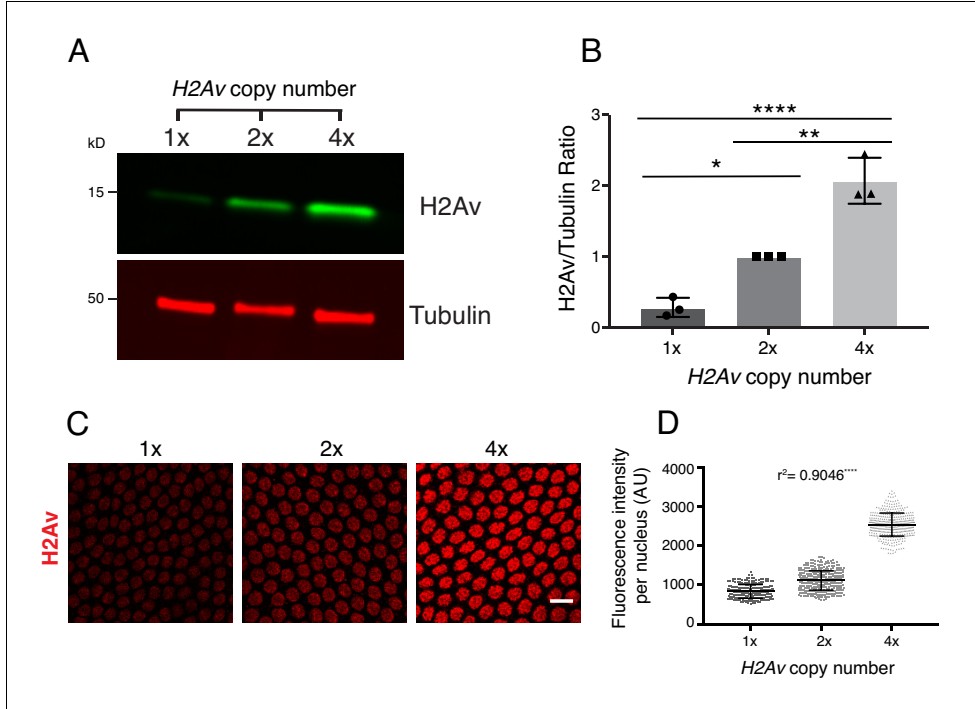

**Figure 8.** Total and nuclear H2Av levels are determined by *H2Av* gene dosage. (**A**) Global H2Av protein levels scale with *H2Av* gene dosage. Total protein from equal numbers of NC14 embryos laid by mothers with either 1, 2, or 4 copies of the *H2Av* gene were separated by SDS PAGE and transferred to membranes. Membranes were probed for H2Av (green) and for tubulin (red), as a loading control. (**B**) Quantitation of (**A**) expressed as the H2Av/tubulin ratio. N = 3. 1x vs. 2x, p=0.0120; 2x vs. 4x, p=0.0016; 1x vs. 4x, p<0.0001. All p values were calculated using one-way ANOVA followed by Tukey's test. (**C**) Anti-H2Av immunostaining of NC13 embryos shows that nuclear H2Av levels scale with *H2Av* gene dosage. (**D**) Quantitation of (**C**) showing total fluorescence intensity within individual nuclei (grey dots) in embryos of varying *H2Av* gene dosage. To test for a relationship between nuclear H2Av levels and H2Av gene dosage, a one-way ANOVA was performed followed by a test for linear trend in Prism (GraphPad). Nuclei within a single embryo were averaged, three embryos were analyzed per replicate, N = 3. $r^2$ = 0.9046. p<0.0001. All error bars represent SD.

DOI: https://doi.org/10.7554/eLife.36021.013

Contrary to our prediction, these FRAP experiments did not show a difference in H2Av-GFP import rates in 0x *Jabba* compared to 1x *Jabba* (*Figure 10E,F*). We propose that this discrepancy is due to the maturation time of eGFP: within the short time scale of our experiments (i.e. several minutes), the most recently synthesized H2Av-GFP will not be visible, as eGFP maturation occurs with a time constant of ~30 min (*Heim et al., 1995*). Thus, our measurements are blind to a fraction of total H2Av. The relative contribution of this 'invisible' fraction will be larger in 0x *Jabba* because here newly synthesized H2Av is not retained in the cytoplasm by LDs and thus does not have a chance to mature before being imported into nuclei. Therefore, measuring recovery of fluorescent H2Av-GFP will underestimate H2Av nuclear import compared to the other genotypes. In the future, it will be possible to test this notion directly by using rapidly maturating fluorescent proteins, such as superfolder GFP.

## H2Av incorporation into nuclei scales with interphase length

Taken together, our quantitative model suggests that the role of LD binding is simply to slow down nuclear import of H2Av, that is that it controls nuclear H2Av levels kinetically. Thus, these levels should go up when interphase is longer, that is with larger t in *Equation (1)* or further to the right in *Figure 9C*. During syncytial blastoderm stages, interphase length increases gradually, from ~10 min in NC10 to ~19 min in NC13 (*Blythe and Wieschaus, 2015a*). We previously noted that in *Jabba* mutant embryos, nuclear H2Av levels increase dramatically during these cycles, paralleling the

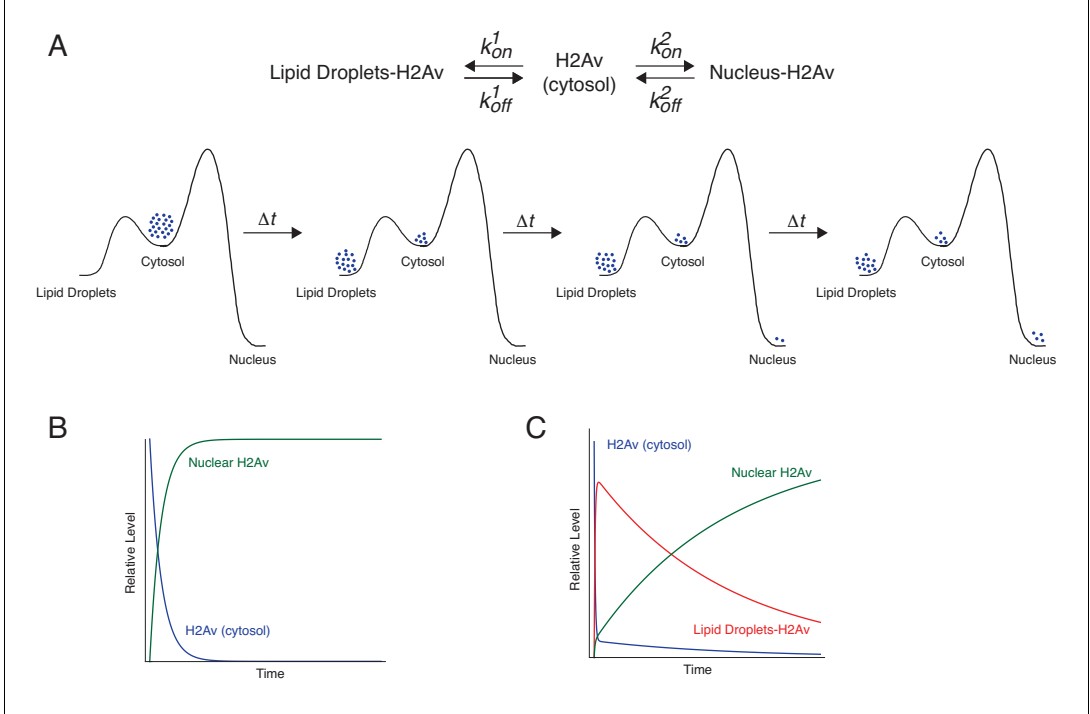

**Figure 9.** Proposed kinetic model for LD-mediated H2Av buffering. (**A**) Proposed kinetic model for the buffering effects of LDs on nuclear incorporation of H2Av. (Top) H2Av free in the cytosol binds LDs with on rate $k^1_{on}$ and off rate $k^1_{off}$ and binds to DNA with on rate $k^2_{on}$ and off rate $k^2_{off}$. (Bottom) With $k^1_{on} > k^2_{on}$, a rapid pre-equilibration of H2Av between the cytoplasm and LDs is established (Left). Over time, however, nuclear H2Av gradually accumulates as its relative energy is lower than H2Av bound to LDs (Right). (**B**) Model showing how free H2Av (Blue) and DNA-bound H2Av (Green) levels change over time in the absence of LDs. (**C**) Model with same parameters as (**B**) showing changes in free H2Av (Blue), DNA-bound H2Av (Green), and LD-bound H2Av (Red) in the presence of LDs. With LDs, nuclear accumulation of H2Av is dramatically slowed (green line in B vs. C). For full derivation of model, see Appendix 1. See also *Figure 9—figure supplement 1*.

DOI: https://doi.org/10.7554/eLife.36021.014

The following figure supplement is available for figure 9:

**Figure supplement 1.** Effects of H2Av levels and *Jabba* dosage on nuclear H2Av accumulation as predicted by the model in *Figure 9*.

DOI: https://doi.org/10.7554/eLife.36021.015

increase in interphase length (*Li et al., 2014*). Furthermore, we find that in the wild type, nuclear H2Av-GFP is increased in subsequent nuclear cycles (*Figure 7A,B*). We therefore measured the rate of H2Av-GFP import into nuclei during these stages, using FRAP. We find that this rate is fairly constant throughout interphase (*Figure 7C*). Thus, lengthening of interphases from NC10 to NC13 can indeed explain the increased nuclear H2Av levels at the ends of these cycles.

To directly test if lengthening interphase increases H2Av accumulation, we injected embryos with the translation inhibitor cycloheximide. Mitotic entry depends on new synthesis of cyclins and thus does not occur if translation is abolished (*McCleland and O'Farrell, 2008*). Using FRAP, we measured the rate of import of H2Av-GFP into nuclei of these embryos. Despite a five-fold increase in interphase length, nuclear H2Av-GFP recovers at similar rates throughout the entire period (*Figure 7C*).

Ongoing H2Av nuclear import throughout a prolonged interphase suggests that H2Av buffering by LDs is regulated through kinetics, rather than absolute H2Av levels, that is the extent of H2Av buffering is not altered once a certain threshold of nuclear H2Av is achieved. So far, we have found no evidence that this import is easily saturated since H2Av levels continue to rise dramatically in cycloheximide injected embryos (*Figure 7C*); thus the conditions of our experiments are still far from reaching the point when nuclear import of H2Av levels off (*Figure 9C*, far right).

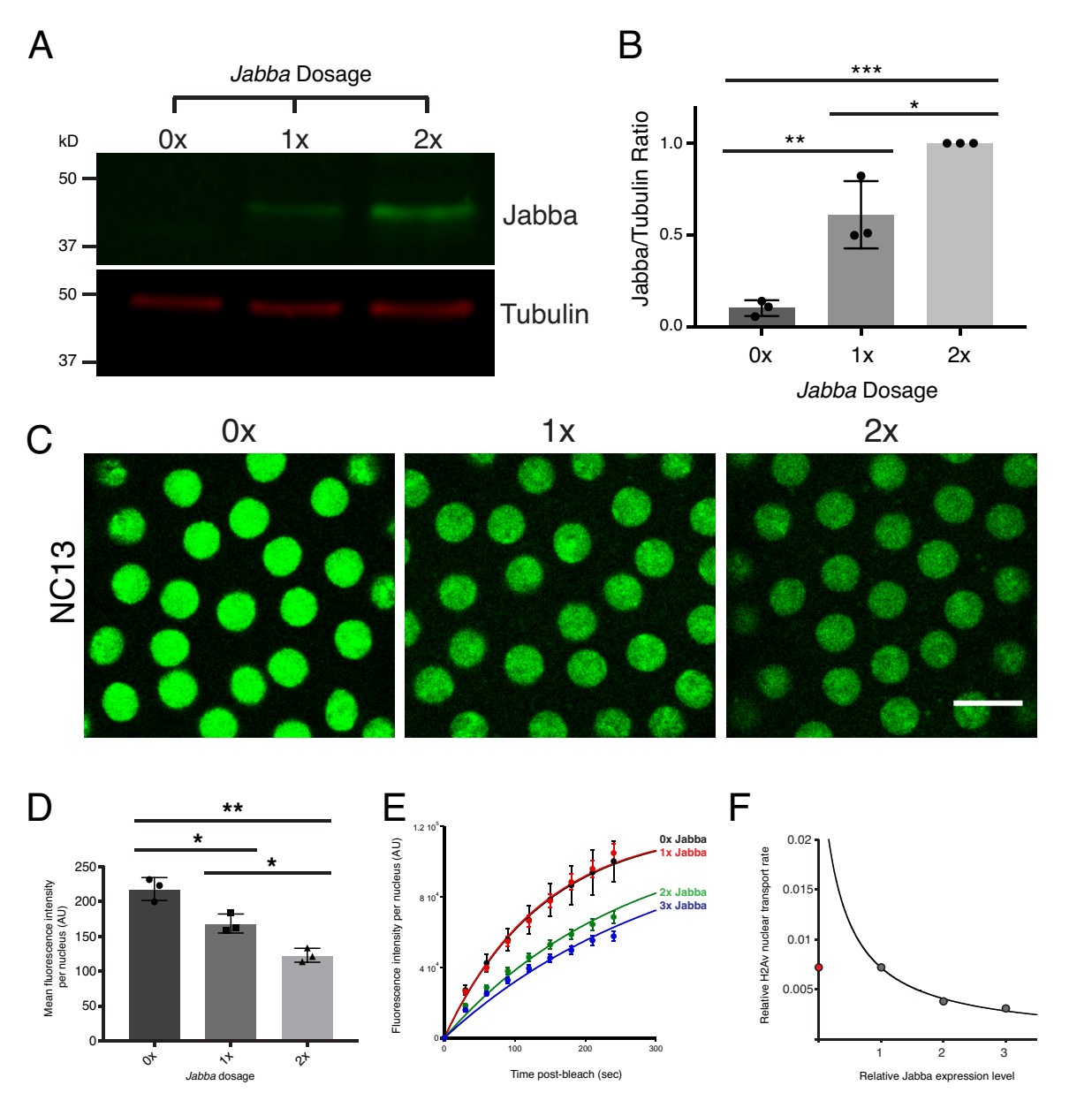

**Figure 10.** Reduction in buffering capacity increases nuclear H2Av. (**A**) Jabba protein levels scale with *Jabba* gene dosage. Total protein from equal numbers of NC14 embryos laid by mothers with either 0, 1, or 2 copies of the *Jabba* gene were separated by SDS PAGE and transferred to membranes. Membranes were probed for Jabba (green) and tubulin (red). (**B**) Quantitation of (**A**) expressed as the Jabba/tubulin ratio. N = 3. 0x vs. 1x, p=0.0030; 1x vs. 2x, p=0.0110; 0x vs. 2x, p=0.0001. All p values were calculated using one-way ANOVA followed by Tukey's test. (**C**) Nuclear H2Av levels inversely scale with *Jabba* dosage. Equally timed NC13 embryos expressing H2Av-GFP and varying copies of *Jabba*. Scale bar represents 10 μm. (**D**) Quantitation of total fluorescence within individual nuclei in NC13 embryos of varying *Jabba* dosage. Five nuclei were averaged per embryo. N = 3 embryos. 0x vs. 1x, p=0.0200; 1x vs. 2x, p=0.0264; 0x vs. 2x, p=0.0018. All p values were calculated using two-way ANOVA followed by Tukey's test. All error bars represent SD. (**E**) FRAP experiments show higher rates of H2Av-GFP nuclear recovery with reduced *Jabba* dosage. FRAP was performed near the surface of NC13 embryos expressing H2Av-GFP and nuclear recovery was monitored over time. Colors represent different *Jabba* dosages. Data points were fit to an exponential curve. (**F**) Rate constants derived from (**E**) plotted as a function of relative Jabba expression levels. The curve is calculated for the predicted relationship between rate constants and fitted for 1x, 2x, and 3x *Jabba* dosage (grey data points). 0x *Jabba* is plotted in red and omitted from curve (see main text).

DOI: https://doi.org/10.7554/eLife.36021.016

## H2Av buffering by LDs is developmentally regulated

Controlling nuclear H2Av levels by slowing import is an elegant way to prevent over-accumulation of H2Av despite massive extra-nuclear supplies. However, its ability to respond to changing conditions is limited, as levels scale with interphase length. If nuclear H2Av were to continue to rise in this fashion throughout the remainder of embryogenesis (~22 hr), there would be enormous increase in H2Av per nucleus relative to other histones, so that the majority of nucleosomes would contain H2Av. *A priori*, this seems unlikely.

To evaluate to what extent our model holds true later in embryogenesis, we probed for potential changes in H2Av nuclear import using FRAP. Remarkably, we found that already by NC14, nuclear import rates of H2Av are drastically reduced compared to NC13 (*Figure 11A,B*). We performed photoswitching experiments as in *Figure 6*, and found no evidence for any substantial loss of H2Av from nuclei (*Figure 11—figure supplement 1*), indicating there is still no substantial nuclear export of H2Av.

Reduced import of H2Av may come about by lower free H2Av levels. We have evidence that two mechanisms synergize to indeed reduce the amount of available H2Av. First, as expected for many other maternal messages, *H2Av* mRNA levels are drastically reduced in NC14 compared to earlier cycles, consistent with previous reports (*Figure 11C*). Second, we found that H2Av exchange between LDs ceases: Photoswitching experiments at various time-points throughout NC14 revealed that while right after mitosis 13 (early NC14) H2Av is similarly dynamic as in earlier stages, by 15 min into cycle 14 (mid-cycle 14) we detected neither loss nor gain of H2Av-Dendra2 on LDs (*Figure 11D–F*, *Figure 11—figure supplement 2*).

## Transition in H2Av dynamics is controlled by the nuclear:cytoplasmic (N:C) ratio

Early in NC14, LDs undergo another dramatic transition: they redistribute away from the embryo periphery and accumulate near the central yolk (*Welte, 2015*). To determine if the two transitions are related, we abolished the inward shift of LDs by eliminating the kinesin co-factor Halo; in *halo* mutant embryos, LDs remain near the embryo surface (*Arora et al., 2016*). Photoswitching experiments in such mutants revealed a similar change in H2Av dynamics as in the wild type (*Figure 11E*). We conclude that the transition in histone dynamics is independent of the intracellular distribution of LDs.

Intriguingly, the radical change in H2Av dynamics occurs at the time of a major developmental transition, when maternal control of embryogenesis ceases and zygotic regulation takes over. This midblastula transition (MBT) marks the transition from maternal to zygotic control of development (*Yuan et al., 2016*). Major events of the MBT include cell-cycle remodeling, large-scale zygotic genome activation, and degradation of a subset of maternal transcripts. To determine whether the transition in H2Av dynamics is a consequence of genome activation, we injected embryos with the RNA Polymerase II inhibitor α-amanitin to abolish new transcription. Inhibition was effective, as judged by lack of membrane invagination and failure of LDs to move away from the periphery, events known to require zygotic transcription (*Arora et al., 2016*; *Zolokar and Erk, 1976*; *Gutzeit, 1980*; *Arking and Parente, 1980*). Photoswitching experiments revealed that the transition still occurred, on a normal time scale (*Figure 11—figure supplement 3*). Thus, the transition is not brought about by newly expressed zygotic genes, but must be due to some maternal input.

Many of the events of the MBT are regulated by one of two biological clocks: elapsed time since fertilization and/or the nuclear:cytoplasmic (N:C) ratio (*Blythe and Wieschaus, 2015b*). The effects of these two clocks can be distinguished by using haploid embryos: these embryos undergo an additional division compared to diploid embryos in order to achieve the same N:C ratio prior to the onset of the MBT (*Edgar et al., 1986*). Therefore, events controlled by the N:C ratio should be delayed in haploid embryos (*Lu et al., 2009*).

We generated haploid embryos by crossing H2Av-Dendra2 females to males homozygous for the recessive male sterile mutation *ms(3)K81* (*Yasuda et al., 1995*) and performed photoswitching experiments to monitor histone dynamics. H2Av continued to exchange rapidly between LDs in early NC15, corresponding to a time after fertilization when exchange has completely stopped in the wild type (*Figure 11G*). Yet, by 15 min into NC15, histones were stably associated with LDs, similar to the situation in NC14 diploid embryos; presumably the ongoing DNA replication at the beginning of

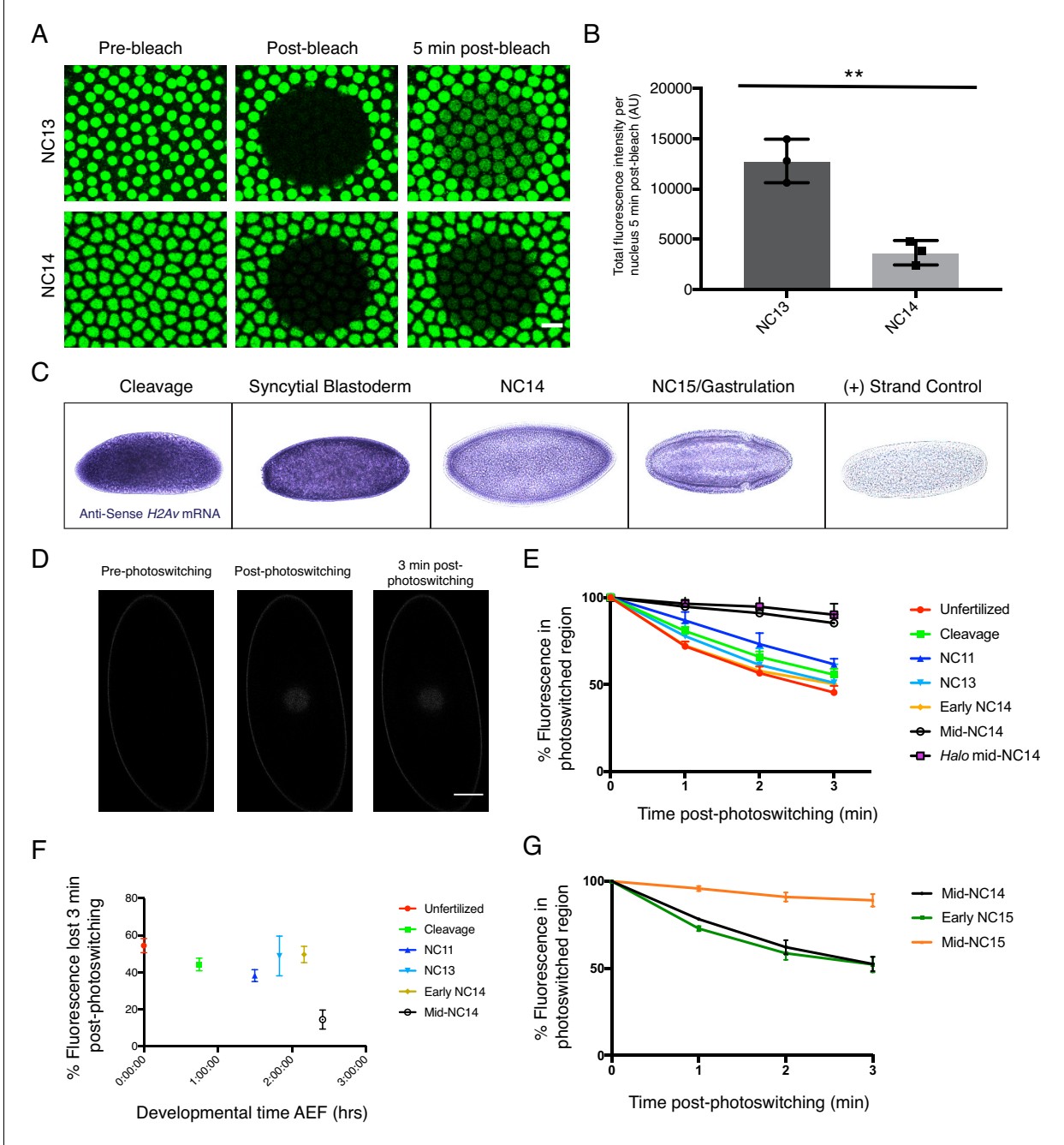

**Figure 11.** H2Av buffering by LDs is developmentally regulated. (**A**) FRAP experiments show reduced nuclear import of H2Av in NC14 compared to NC13. FRAP was performed near the surface of NC13 and NC14 embryos expressing H2Av-GFP, and nuclear recovery was monitored over time. Scale bar represents 10 μm. (**B**) Quantitation of (**A**). Total fluorescence intensity per nucleus ~5 min post-bleaching was measured. Ten nuclei were averaged per embryo. N = 3 embryos. NC13 vs. NC14, p=0.0031. The p value was calculated using an unpaired student t-test. See also *Figure 11—figure supplement 1*. (**C**) In situ hybridization shows decreased *H2Av* mRNA levels at NC14. Representative images of embryos probed with anti-sense *H2Av* mRNA (purple). As a control, embryos were also probed with sense *H2Av* mRNA (far right). (**D**) H2Av-Dendra2 loss from LDs is dramatically reduced in mid-NC14 embryos. Photoswitching was induced in mid-NC14 embryos as in *Figure 3B* and fluorescent signal within the photoswitched region was monitored over time (see also *Figure 11—figure supplement 2*). (**E**) Representation of data from (**D**) and *Figure 3C* showing the percentage of fluorescence signal remaining within the photoswitched region over time in the indicated stages/genotypes. N = 5 embryos per stage. (**F**) Data from (**E**) plotted as total percentage of fluorescent signal lost from the photoswitched region ~3 min post-photoswitching across early development. AEF = After Egg Fertilization. (**G**) Transition in H2Av buffering is determined by the nuclear:cytoplasmic (N:C) ratio. Photoswitching experiments were performed in haploid embryos. Quantitation of the percent fluorescence in the photoswitched region reveals that the transition in H2Av buffering is delayed until mid-NC15 in haploid embryos. N = 3 embryos per stage. All error bars represent SD. See also *Figure 11—figure supplement 3*.

*Figure 11 continued on next page*

*Figure 11 continued*

DOI: https://doi.org/10.7554/eLife.36021.017

The following figure supplements are available for figure 11:

**Figure supplement 1.** Nuclear export of H2Av is minimal in NC14.

DOI: https://doi.org/10.7554/eLife.36021.018

**Figure supplement 2.** H2Av-Dendra2 loss from LDs is reduced in mid-NC14.

DOI: https://doi.org/10.7554/eLife.36021.019

**Figure supplement 3.** Zygotic transcription is not required for transition in H2Av dynamics.

DOI: https://doi.org/10.7554/eLife.36021.020

these cycles pushes the N:C ratio above a threshold level to switch histone dynamics. Thus, the transition in histone dynamics is not controlled by developmental time, but rather by the N:C ratio. We conclude that the transition in the dynamics of H2Av association with LDs is a previously unrecognized part of the MBT.

## Discussion

### Lipid droplets are the main regulator of nuclear H2Av in early embryos

Assuring that chromatin contains the correct amount of histone proteins requires multiple steps, from production to chromatin deposition (*Figure 5—figure supplement 1*). For most histone species, histone incorporation is regulated by the abundance of histone proteins, via control of transcription, mRNA turnover, and protein degradation (*Marzluff and Duronio, 2002*; *Gunjan and Verreault, 2003*). Here, we provide evidence that for the variant histone H2Av in early *Drosophila* embryos the dominant step for regulating histone incorporation is not histone abundance per se, but post-translational regulation of available H2Av protein via buffering by LDs: in these embryos, LDs transiently sequester excess H2Av, effectively depleting the H2Av pool that is readily available for nuclear import and chromatin deposition (*Figure 5A*). Varying buffering capacity – via *Jabba* dosage changes – indeed modulates H2Av nuclear import rates and controls total nuclear H2Av levels (*Figure 10*).

We hypothesize that such a simplified system for histone regulation represents an adaptation necessary for controlling nuclear histone levels during periods of extremely rapid nuclear proliferation. During early *Drosophila* embryogenesis, a modified cell-cycle program allows for a nuclear division every 8 min. Such rapid cell cycles may make it challenging to regulate histone levels via canonical pathways. In budding yeast, for example, non-chromatin bound histones are degraded with a half-life of ~30 min (*Singh et al., 2010*). This hypothesis is consistent with our findings that buffering is turned off in NC14, when a prolonged interphase, including the first gap phase, is introduced. In addition, some of the canonical pathways are simply not available: for the first ~2 hr, the zygotic genome is largely inactive, and thus transcription could at most make a minor contribution to *H2Av* mRNA abundance. Histone mRNAs are instead provided maternally and if they were degraded in early cycles, they would no longer be available later; indeed, they are largely stable until NC14. Abundant maternally provided H2Av protein is supplemented by newly synthesized H2Av translated from maternal *H2Av* mRNAs (*Li et al., 2014*); however, there does not seem to be any feedback regulation to adjust levels to usage, as H2Av proteins scale with *H2Av* gene dosage, suggesting that neither translation nor degradation are tuned to achieve a particular H2Av abundance.

We found no evidence for substantial export or loss of H2Av from nuclei during the time frame of our experiments (*Figure 6C,D*, *Figure 11—figure supplement 1*), and thus H2Av nuclear levels are indeed dominated by import (*Figure 5A*). We speculate that H2Av imported into the nucleus quickly becomes chromatin-bound and is therefore stably retained. Although our experiments do not directly address chromatin association during interphase, at least in mitosis the vast majority of H2Av signal appears to localize to chromosomes (*Figure 7A*). Because H2Av is typically present in chromatin ten-fold less abundantly than H2A (*Leach et al., 2000*), there should be lots of potential sites where H2Av could be incorporated into nucleosomes instead of H2A. Indeed, during an artificially elongated interphase (*Figure 7C*), nuclear H2Av continues to rise at a similar rate throughout, suggesting that H2Av incorporation is far from saturation.

Our data uncovers a role for LDs as H2Av buffers; at the moment, it remains unresolved whether other LD-associated histones (i.e. H2A and H2B) are regulated similarly. We hypothesize that buffering is limited to H2Av: previous experiments have shown that nuclear H2A levels are not obviously altered in *Jabba* mutants (*Li et al., 2014*). As H2B is vastly more abundant on LDs than H2Av (*Cermelli et al., 2006*; *Li et al., 2014*), most of it can in principle be regulated independently of H2Av, even though H2Av is presumably present on LDs as an H2Av-H2B heterodimer. Core histones are generally regulated quite differently than histone variants, so the fact that differences exist in early embryos would not be surprising. Perhaps, the two functions of LDs in histone regulation are mutually exclusive: LDs store core histones, H2A and H2B, for later use (i.e. histone storage function [*Li et al., 2012*]) and dynamically sequester H2Av to reduce availability (i.e. histone buffering function). Future live-imaging of the core histones will be necessary to address this question.

## Jabba/LDs as a novel histone chaperone

Histone chaperones are broadly defined as a group of proteins that both bind histones and regulate nucleosome assembly (*Burgess and Zhang, 2013*). The ability of LDs to transiently sequester histones via the anchor protein Jabba, and its subsequent role in limiting histone deposition onto chromatin, argues that LDs have a previously uncharacterized role as histone chaperones. The function of LDs in regulating free H2Av levels is remarkably similar to other known histone chaperones. The H3-H4 histone chaperone nuclear autoantigenic sperm protein (NASP), for example, has been shown to regulate soluble H3-H4 levels in HeLa cells: non-chromatin bound H3-H4 pools are lost in NASP mutants, and NASP overexpression is sufficient to increase soluble H3 and H4 levels (*Cook et al., 2011*). Additionally, in *Xenopus*, the H2A/H2B chaperone nucleoplasmin (Npm) has been implicated in both histone storage and release, a functional transition that has been shown to be developmentally regulated (*Onikubo et al., 2015*). Despite such similarities, however, our data reveals a key difference: histone storage in *Drosophila* embryos is highly dynamic while, in *Xenopus*, storage is largely static. We hypothesize that the dynamic nature of histone storage in *Drosophila* represents a necessary adaptation to cope with the rapid cell cycles, which are ~3 x shorter than in *Xenopus*. Additionally, *Xenopus* cleavage divisions do not occur within a syncytium: therefore, dynamic buffering by LDs would, presumably, not be as robust, as each division would split the maternal H2Av pool. It is also possible, as alluded to above, that core histones are statically stored, while dynamic storage is employed for the histone variants.

## Quantitative modeling of histone import

Our quantitative model of how LDs modulate the nuclear accumulation of H2Av (*Figure 9*, Appendix 1) reproduces key aspects of the behavior observed in vivo (*Figures 7*,*8*,*10*). This correspondence is remarkable, given that the model takes into account just a few molecular players (*Figure 5A*); this suggests that many other regulatory mechanisms known to affect histone accumulation (*Figure 5—figure supplement 1*) play only limited roles in the early *Drosophila* embryo. The next steps will be to refine this quantification in the future, by directly measuring the rate constants in *Figure 9A* as well as the absolute concentrations of various histone pools. Together with resolving the contribution of not fully matured, 'invisible' H2Av-GFP, this will set the stage for defining if other molecular players need to be considered to fully account for the observed import rates. For example, the current model does not yet incorporate the modest, but detectable, increase in total H2Av levels over the course of cleavage and syncytial blastoderm stages (*Li et al., 2014*). On the minute time frame of individual import experiments, this rise due to new translation is likely irrelevant, but it might make a contribution to how import changes between nuclear cycles.

For simplicity, our model assumes that incorporation of histones into DNA is driven by the concentration of H2Av in the free cytoplasmic pool and the number of H2Av-binding sites on DNA (*Figure 9A*, top, right). However, in principle, any other aspect of H2Av assembly onto chromatin might be the relevant step. We already have an indication that the concentration of H2Av-binding sites on DNA ([*D*]) does not influence the rate of formation of DNA-bound H2Av, because – when we compare across nuclear cycles – the rate of loss of H2Av from LDs is independent of the number of nuclei (*Figure 3C*). Thus, the rate of H2Av incorporation may not be limited by its interaction with DNA, but rather by its interactions with auxiliary factors such as histone chaperones that regulate nucleosome assembly. Like many other maternal proteins, histones chaperones are likely to present

at constant levels in the early embryos. Systematic dosage changes for known chaperones should reveal whether any of these factors is rate-limiting in early embryos.

## Transition in H2Av buffering by LDs represents a previously unrecognized aspect of the mid-blastula transition (MBT)

The dramatic transition in H2Av dynamics observed during NC14 (*Figure 11D–F*) suggests that abolishing H2Av buffering may be an import aspect of the MBT. We hypothesize that this change is necessary to allow for canonical regulatory mechanisms to take over. Static sequestration of histones to LDs will immobilize the large pool of previously generated H2Av and thus prevent over-accumulation in nuclei during the extended interphase of NC14. Overabundance of histones in general can cause cytotoxicity, via multiple mechanisms (*Singh et al., 2010*); and over-accumulation of specifically H2Av in *Drosophila* embryos results in activation of a DNA damage pathway and reduced hatching (*Li et al., 2014*). Later, the dramatic reduction in *H2Av* mRNA levels presumably limits new H2Av protein biosynthesis; if LDs were still able to bind H2Av and act as a sink, this might reduce effective H2Av concentrations below critical levels; dearth of histones is known to severely disrupt chromatin function (*Han et al., 1987*).

## Lipid droplets as general sites of protein sequestration/regulation

It has long been known that proteins from a myriad of cellular compartments can be recruited to LDs (*Welte, 2007*; *Welte, 2015*). In particular, there is a growing list of instances where proteins partition between LDs and the nucleus in a controlled manner. First, histones are present on LDs not only in *Drosophila* embryos, but also in many other cells, including in mammalian systems (*Welte, 2015*). For example, citrullinated H3 is present on LDs in early mouse oocytes (*Kan et al., 2012*); and H2A and H2B, were identified as high confidence LD proteins in human osteosarcoma (U2OS) and human hepatocellular carcinoma (Huh7) cell lines (*Bersuker et al., 2018*). Whether in these cases LD localization also modulates histone availability for chromatin assembly remains to be elucidated. Second, the Perilipin PLIN5 has dual roles at the LD surface and in the nucleus: at the LD surface it modulates lipase activity to regulate triglyceride breakdown while in the nucleus it increases expression of genes that promote mitochondrial biogenesis (*Gallardo-Montejano et al., 2016*). These two PLIN5 populations are connected: under well-fed conditions, all PLIN5 is present on LDs; under starvation conditions, a fraction of PLIN5 relocates to the nucleus to act as a transcriptional co-factor and the fraction that remains LD-associated promotes lipolysis. In that manner, a single protein coordinates both the generation of fatty acids from triglycerides and their efficient use by mitochondria. Third, the transcription factor NFAT5 can be sequestered on LDs, via tethering to the LD protein Fsp27/CIDEC; it has been proposed that this sequestration prevents its translocation to the nucleus and thus regulates gene expression (*Ueno et al., 2013*). Finally, CCT1 (Choline-phosphate cytidylyl transferase 1), a key enzyme in phospholipid metabolism is present at the surface of LDs during times of triglyceride synthesis and is critical for LD expansion (*Krahmer et al., 2011*); however, under basal conditions, it relocates to the nucleus (although it is unclear if nuclear accumulation is functionally important).

These observations are part of the emerging evidence that – beyond their role in lipid metabolism – LDs have a broad function in protein handling for proteins destined for all kinds of compartments. Previously characterized roles include the maturation, sequestration, and turnover of select proteins (*Welte and Gould, 2017*). Our studies now reveal that sequestration can be dynamic, rather than static, and that this can result in a buffering role.

## Lipid droplets as regulators of development

Our findings strongly suggest that LDs can modulate developmental progression: prior to the MBT, dynamic association of H2Av with LDs allows for rapid, but restrained nuclear import. This dynamic state quickly transitions to static sequestration during NC14, contributing to a dramatically reduced rate of nuclear import (*Figure 11A,B*). The ability of LDs to differentially modulate H2Av import suggests an important function as regulators of embryogenesis. Of course, LDs can have profound effects on development as the source of hydrophobic signaling molecules (*Welte and Gould, 2017*). But to our knowledge, this is the first example of LDs promoting proper developmental progression via their protein-handling function. Given that LDs are nearly ubiquitous and often are particularly

abundant in oocytes and embryos, their potential as novel mediators of developmental regulation is vast. The tools to test this idea will become increasingly available for many experimental systems since our understanding of the basic cell biology and physiology of LDs is growing at an ever-increasing rate (*Welte and Gould, 2017*; *Walther et al., 2017*; *Welte, 2015*; *Gross and Silver, 2014*; *Kory et al., 2016*; *Murphy, 2012*; *Thiam et al., 2013*; *Coleman and Hesselink, 2017*).

# Materials and methods

**Key resources table**

| Reagent type | Designation | Source or reference | Identifiers | Additional information |
|---|---|---|---|---|
| Gene (*D. melanogaster*) | *His2Av* | NA | FBgn0001197 | |
| Gene (*D. melanogaster*) | *Jabba* | NA | FBgn0259682 | |
| Gene (*D. melanogaster*) | *halo* | NA | FBgn0001174 | |
| Gene (*D. melanogaster*) | *LSD-2* | NA | FBgn0030608 | |
| Genetic reagent (*D. melanogaster*) | *H2Av810* | Bloomington Drosophila Stock Center | BDSC:9264; FLYB: FBst0009264 | |
| Genetic reagent (*D. melanogaster*) | *H2Av-GFP* | Bloomington Drosophila Stock Center | BDSC:24163; FLYB: FBst0024163 | |
| Genetic reagent (*D. melanogaster*) | *H2Av-mRFP* | Bloomington Drosophila Stock Center | BDSC:23650; FLYB: FBst0023650 | |
| Genetic reagent (*D. melanogaster*) | *ms(3)K81* | Bloomington Drosophila Stock Center | BDSC:53252; FLYB: FBst0005352 | |
| Genetic reagent (*D. melanogaster*) | *H2Av-paGFP* | other | FLYB: FBtp0020089 | Described in (*Post et al., 2005*) |
| Genetic reagent (*D. melanogaster*) | *LSD2-YFP* | Kyoto Stock Center | DGRC:115301; FLYB: FBti0143786 | |
| Genetic reagent (*D. melanogaster*) | *JabbaDL*, *Jabbazl01* | other | FLYB: FBal0280317; FBal0280318 | Previously generated *Jabba* null alleles. Described in (*Li et al., 2012*). |
| Genetic reagent (*D. melanogaster*) | *Df(2L)ΔhaloAJ* | other | FLYB: FBab0047638 | Small deletion encompassing the *halo* gene. Described in (*Arora et al., 2016*). |
| Genetic reagent (*D. melanogaster*) | *H2Av-Dendra2* | this paper | | genomic H2Av region (~4 kb), Dendra2 inserted downstream of exon 4, cloned into pattB, genomic insertion site 68A4 |
| Genetic reagent (*D. melanogaster*) | *gH2Av* | this paper | | genomic H2Av region (~4 kb), cloned into pattB, genomic insertion site 68A4 |
| genetic reagent (*D. melanogaster*) | *gJabba* | this paper | | genomic Jabba region (~5.3 kb). Insertion site 68A4 |
| Antibody | anti-H2AvD (rabbit polyclonal) | Active Motif | Cat. No. 39715 | (1:1000 immunostain) (1:2500 WB) |
| Antibody | anti-Jabba (rabbit polyclonal) | this paper | | raised against in-vitro synthesized peptide encoded in exon 5 of Jabba followed by affinity purification |
| Antibody | anti-alpha tubulin (mouse monoclonal) | Cell Signaling | Cat. No. #3873 | (1:10,000) |
| Antibody | IRDye secondaries 800CW or 680RD | Li-COR | | (1:10,000) |
| Antibody | Alexa 594 secondary | ThermoFisher | Cat. No. A-11012 | (1:1000) |
| Commercial kit | mMESSAGE mMACHINE T7 | Ambion Inc. | Cat. No. AM1344 (Fisherscientific) | |

*Continued on next page*

*Continued*

| Reagent type | Designation | Source or reference | Identifiers | Additional information |
|---|---|---|---|---|
| Drug | α-amanitin | Sigma | Cat. No. A2263 | (500 µg/mL) |
| Drug | cycloheximide | Sigma | Cat. No. C7698 | (1 mg/mL) |

## Fly stocks

Oregon R was used as the wild-type stock. Lines carrying the following mutations were obtained from the Bloomington *Drosophila* Stock center: $H2Av^{810}$(*van Daal and Elgin, 1992*), H2Av-GFP (*Clarkson and Saint, 1999*), H2Av-mRFP (*Schuh et al., 2007*), and ms(3)K81 (*Yasuda et al., 1995*). *halo* (*Arora et al., 2016*) and H2Av-paGFP (*Post et al., 2005*) stocks were previously described. The LSD2-YFP stock (*Lowe et al., 2014*) was obtained from the Kyoto Stock Center. The H2Av-paGFP line (*Post et al., 2005*) was a gift from Eric Wieschaus. To analyze embryos lacking *Jabba*, we employed either of two null alleles, $Jabba^{zl01}$ and $Jabba^{DL}$ (*Li et al., 2012*).

To obtain H2Av-GFP-expressing embryos with varying *Jabba* dosage, females of the following genotypes were used: $Jabba^{DL}/Jabba^{DL}$, H2Av-GFP/TM3 (0x *Jabba*), $Jabba^{DL}/CyO$, H2Av-GFP/TM3 (1x *Jabba*), +/+, H2Av-GFP/TM3 (2x *Jabba*), or +/+, gJabba/H2Av-GFP (3x *Jabba*, transgene generation described below).

To analyze embryos with varying *H2Av* dosage, mothers with the following genotypes were used: $H2Av^{810}/TM3$ (1x *H2Av*), *Oregon-R* (2x *H2Av*), or flies homozygous for the *H2Av* transgene *gH2Av* (4x *H2Av*), described below, in an otherwise wild-type background.

Unless otherwise indicated, all fly lines expressing H2Av fusion proteins (e.g. H2Av-Dendra2, H2Av-GFP) are wild type at the endogenous *H2Av* locus. As such, total H2Av levels are likely slightly elevated in these experiments (*Figure 8*). Importantly, however, all experiments in which kinetics were compared used the same H2Av-fusion protein and have the same ratio of tagged:untagged H2Av.

## Transgene generation

To generate a genomic *H2Av* transgene (*gH2Av*), a ~4 kb genomic region encompassing *H2Av* (including the promoter region and 3'UTR) was TOPO cloned into the pCR4Blunt-TOPO vector (Invitrogen, Carlsbad, CA). This genomic region is almost identical to the one used to generate the widely employed *H2Av-GFP* gene (*Clarkson and Saint, 1999*). For *H2Av-Dendra2*, the Dendra2 protein coding sequence (Evrogen, Moscow, RUS) was inserted immediately downstream of the fourth *H2Av* exon, prior to the stop codon, via standard cloning techniques. To generate a genomic *Jabba* transgene (*gJabba*), a ~5.3 kb genomic region encompassing *Jabba* (including the promoter region and 3'UTR) was TOPO cloned into the pCR4Blunt-TOPO vector (Invitrogen).

All constructs (*gH2Av*, *H2Av-Dendra2*, and *gJabba*) were then isolated via PCR and ligated into the pattB plasmid. Transgenic fly lines were created via PhiC31 integrase-mediated transgenesis, and transgenes were incorporated onto the third chromosome, site 68A4 (BestGene Inc., Chino Hills, CA).

## In situ hybridization

*H2Av* RNA probes were generated in vitro using mMESSAGE mMACHINE T7 transcription kit (Ambion Inc., Foster City, CA). Embryos were formaldehyde-fixed and prepared as described previously (*Wilk et al., 2010*). RNA probes were labelled with digoxigenenin and detected using NBT/BCIP substrate (Roche, Basel, CHE). Bright-field images were acquired using a Nikon Eclipse E600 fluorescence microscope with a 4MP Spot Insight camera.

## Embryo injections

Embryos were dechorionated by hand by rolling on double-stick tape, dehydrated for 7–10 min depending on environmental conditions, and overlayed with Halocarbon oil 700 (Sigma, St. Louis, MO). Embryos were injected with either cycloheximide (Sigma) diluted in 5 mM KCl and 0.1 mM $NaPO_4$ (pH 6.5) at a needle concentration of 1 mg/ml or with α-amanitin (Sigma) diluted in water at a needle concentration of 500 µg/mL. Imaging was performed within minutes of cycloheximide

injection. For α-amanitin, the drug was injected in early syncytial blastoderm embryos (NC10-NC11), and the embryos were imaged once they reached NC14.

## Microscopy/Live Imaging

All imagings, unless otherwise indicated, were performed on a Leica SP5 laser scanning confocal microscope with Leica HyD hybrid detectors.

For photoactivation experiments, H2Av-paGFP embryos were dechorionated by hand by rolling on double-stick tape and overlayed with Halocarbon Oil 27 (Sigma) before imaging. Photoactivation was achieved by zooming in on a presumptive LD enriched region (ROI) and exposing to 10 frames of 405 nm light (5.184 s/frame). Imaging was performed using 488 nm excitation (40x objective).

For photoswitching experiments, H2Av-Dendra2 embryos were dechorionated and mounted as previously described. 488 nm excitation (green channel) was used to assess age of embryos (based on nuclear density) and location of LDs and/or nuclei. Photoswitching was performed using the Leica FRAP wizard software (40x objective). Photoswitching was achieved either by zooming in on an ROI and exposing to a single frame of 405 nm light (5.184 s/frame) (e.g. *Figures 3B,4E*) or by using a single 405 nm bleach point (100 ms exposure) (e.g. *Figure 6C*). Imaging of H2Av-Dendra2 was performed using 488 nm excitation (non-photoswitched H2Av-Dendra2-green) or 543 nm excitation (photoswitched H2Av-Dendra2-red). In vivo centrifugation of live embryos was performed as previously described (*Tran and Welte, 2010*).

For FRAP experiments, embryos were dechorionated and mounted as described above. Bleaching was performed by zooming in on a region of interest (ROI) and exposing to 10 frames of 488 nm light (5.184 s/frame) using the Leica FRAP wizard software (40x objective). To ensure accurate staging for FRAP experiments with varying *Jabba* dosages, bleaching was performed 6 min following the first signs of metaphase in NC12.

To measure nuclear H2Av-GFP levels in embryos of various *Jabba* dosages, embryos were dechorionated by hand, placed on a coverslip containing a spot of heptane/double stick tape glue, and overlayed with halocarbon oil. To ensure accurate staging across replicates/genotypes, all images were acquired ~3 min following the NC12 anaphase.

## Immunostaining

Embryos of varying *H2Av* dosage were obtained as described above. Embryos were collected on apple juice agar plates, aged appropriately, dechorionated in 50% bleach, and fixed in 4% formaldehyde/1xPBS for 20 min. Embryos were then de-vitellinized using heptane/methanol and washed extensively in 1x PBS/0.1% Triton X-100 (Sigma). Embryos (equal number of embryos per tube) were incubated in rabbit anti-H2AvD antibody (Active Motif, Carlsbad, CA) at 1:1000 final concentration O/N at 4°C. Embryos were washed extensively in 1x PBS/0.1% Triton X-100 and incubated for 4 hr at RT in goat anti-rabbit Alexa 594 (Invitrogen) at a 1:1000 final concentration. Embryos were washed extensively and mounted in Vectashield (Vectorlabs, Burlington, ON, CAN) for imaging.

## Western analysis/antibody generation

To detect Jabba, H2Av, and Tubulin protein levels, embryos were heat-fixed in 1x TSS, sorted by age, and boiled in 2x Laemmli buffer (Bio-RAD, Hercules, CA). Proteins were separated using 4–15% SDS-PAGE gels (Bio-RAD) and transferred to Immobilon-FL PVDF membranes (MilliporeSigma, Burlington, MA) in CAPS buffer (Jabba) or Towbin buffer (H2Av). Immunodetection was performed using the following primary antibodies O/N at 4°C – rabbit anti-Jabba (1:5000), rabbit anti-H2Av (1:2500) (Active Motif) and mouse anti-α-Tubulin (1:10,000) (Cell Signaling, Danvers, MA). Membranes were washed in 1xPBS/0.1% Tween-20 (Sigma) and incubated in the following secondary antibodies for 1 hr at R/T – IRDye 800CW goat anti-rabbit IgG and IRDye 680RD goat anti-mouse IgG (1:10,000) (Li-COR, Lincoln, NE). Membranes were imaged using Li-COR Odyssey CLx. Bands were quantified using ImageJ.

The rabbit anti-Jabba antibody used was generated via injection of an in vitro synthesized peptide encoded in exon 5 of Jabba (VHKAEEKENDKGK) followed by affinity purification (GeneScript, Piscataway, NJ). This antibody detects only a subset of the bands recognized by the anti-Jabba antibody previously used (*Li et al., 2012*). As the two antibodies were generated against epitopes encoded by different exons, this observation supports our previous speculation that Jabba protein

can be processed into multiple forms (*Li et al., 2012*); this idea will be explored in detail elsewhere. Such processing is not relevant for the analysis here, as we are only comparing embryos with different dosage of wild-type protein. The Western in *Figure 10A* demonstrates the specificity of detection. For quantitation in *Figure 10B*, the most prominent band (shown) was measured.

### Quantitation and image analysis

Quantitation of fluorescent images was performed using ImageJ. For quantitation of signal in individual nuclei, nuclear boundaries were identified via thresholding and watershedding. For measuring H2Av nuclear levels via immunostaining, only nuclei within a certain size range were included in the quantitation: this was to account for potential variations in both focal plane depth and NC13 timing between embryos (i.e. nuclei of similar size indicate a similar focal plane and a similar stage of NC13). For quantitation of fluorescence recovery in *Figure 4D*, total fluorescence lost during bleaching was set to 100%. To account for photobleaching during subsequent imaging, fluorescence intensity was quantified in an equally sized ROI outside of the bleached region and used to normalize fluorescence recovery within the bleached ROI. For these experiments, embryos that exhibited substantial cytoplasmic streaming (as in *Figure 4—figure supplement 1*) were not analyzed. Statistics were performed using an unpaired student t-test or ANOVA followed by a Tukey test or linear regression analysis (Prism7, GraphPad). All graphs were assembled using either Prism 7 or Mathematica 11 (Wolfram). All images were processed in Adobe Photoshop and assembled with Adobe Illustrator.

## Acknowledgements

We thank Erick Wieschaus, the Bloomington Stock Center, and the Kyoto Stock Center for fly stocks. We thank Dan Bergstralh for his upgrades to the departmental confocal microscope. We are grateful to Shelby Blythe, Eric Wieschaus, Bill Sullivan, Jeffrey Hayes, Patrick Oakes, and Danielle Presgraves for advice and discussion and to Arno Müller and Mathias Beller for comments on the manuscript.

## Additional information

### Funding

| Funder | Grant reference number | Author |
|---|---|---|
| National Institutes of Health | RO1 GM102155 | Michael Andreas Welte |

The funders had no role in study design, data collection and interpretation, or the decision to submit the work for publication.

### Author contributions

Matthew Richard Johnson, Conceptualization, Formal analysis, Investigation, Visualization, Methodology, Writing—original draft, Writing—review and editing; Roxan Amanda Stephenson, Investigation, Writing—review and editing; Sina Ghaemmaghami, Formal analysis, Visualization; Michael Andreas Welte, Conceptualization, Supervision, Funding acquisition, Writing—original draft, Writing—review and editing

### Author ORCIDs

Matthew Richard Johnson http://orcid.org/0000-0002-9369-0084
Michael Andreas Welte http://orcid.org/0000-0001-5741-4720

### Decision letter and Author response
Decision letter https://doi.org/10.7554/eLife.36021.024
Author response https://doi.org/10.7554/eLife.36021.025

## Additional files

### Supplementary files
• Transparent reporting form
DOI: https://doi.org/10.7554/eLife.36021.021

### Data availability
All data generated or analyzed during this study are included in the manuscript.

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

## Appendix 1

DOI: https://doi.org/10.7554/eLife.36021.022

To devise a formal kinetic model, we considered the following rate equations describing changes in concentrations of free, LD-bound, and DNA-bound H2Av as a function of time:

$$\frac{dH}{dt} = -k_{on}^1 \cdot [L] \cdot [H] + k_{off}^1 \cdot [HL] - k_{on}^2 \cdot [D] \cdot [H] + k_{off}^2 \cdot [HD] \tag{1}$$

$$\frac{dHL}{dt} = k_{on}^1 \cdot [L] \cdot [H] - k_{off}^1 \cdot [HL] \tag{2}$$

$$\frac{dHD}{dt} = k_{on}^2 \cdot [D] \cdot [H] - k_{off}^2 \cdot [HD] \tag{3}$$

[H] = concentration of unbound, cytosolic H2Av
[D] = concentration of DNA binding sites for H2Av
[L] = concentration of LD binding sites for H2Av
[HD] = concentration of DNA-bound H2Av
[HL] = concentration of LD-bound H2Av $k_{on}^1$, $k_{off}^1$, $k_{on}^2$, and $k_{off}^2$ are defined in **Figure 9A**.
The following simplifying assumptions were made:

(A) The number of potential H2Av binding sites on both DNA and LDs are in excess of free H2Av levels: therefore, [D] and [L] do not appreciably change over the time course of our experiments. These assumptions give rise to the following equations:

$$\frac{dD}{dT} = 0 \tag{4}$$

$$\frac{dL}{dT} = 0 \tag{5}$$

(B) The rate constant of H2Av binding to LDs is greater than that of H2Av binding to DNA (*i.e.* $k_{on}^1 \gg k_{on}^2$). This assumption is based on the observation that H2Av lost from LDs (and thus having passed through the cytoplasmic pool) efficiently accumulates on LDs again (**Figure 4G′**) rather than traveling to nuclei. This assumption means that there is a rapid, pre-equilibrium of H2Av partitioning between the cytosol and LDs.

(C) H2Av bound to DNA is significantly lower in energy than when free in the cytoplasm and, therefore, $k_{on}^2 \gg k_{off}^2$. This assumption is supported by the observation that nuclear export of H2Av is minimal compared to import (see **Figure 6C,D**).

(D) H2Av bound to LDs is significantly lower in energy than when free in the cytoplasm and, therefore, $k_{on}^1 \gg k_{off}^1$, which is equivalent to the equilibrium constant $K_{HL} = \frac{[H] \cdot [L]}{[HL]} = \frac{k_{off}^1}{k_{on}^1} \ll 1$. This assumption is supported by the fact that there is net flux from LDs to nuclei (Fig. 1), but not in the reverse direction (**Figure 6D**).

(E) Total H2Av (H$_{tot}$) is constant over time (*i.e.* there is neither new synthesis nor degradation). This assumption is a close approximation for the time frames considered.

Solving the differential rate equations above and incorporating the simplifying assumptions allowed us to derive an equation (**Equation 6**) representing the formation of DNA-bound H2Av (HD) as a function of time.

$$HD(t) = [H_{tot}] \cdot \left( 1 - e^{\frac{K_{HL}}{K_{HL}+[L]} \cdot [D] \cdot K_{on}^2 \cdot t} \right) \tag{6}$$

This equation relates the rate of formation of DNA-bound H2Av, *HD(t)*, to total H2Av concentration, [$H_{tot}$], the concentration of H2Av binding sites on LDs, [LD], and the concentration of H2Av binding sites on DNA, [D].

