## [Decision Letter]

Thank you for submitting your article "Developmentally regulated H2Av buffering via dynamic sequestration to lipid droplets in *Drosophila* embryos" for consideration by *eLife*. Your article has been reviewed by three peer reviewers, and overseen by a Reviewing Editor and Didier Stainier as the Senior Editor. The following individuals involved in review of your submission have agreed to reveal their identity: Joel Goodman (Reviewer #3).

The reviewers have discussed the reviews with one another and the Reviewing Editor has drafted this decision to help you prepare a revised submission.

Summary:

All reviewers agreed that this is an excellent paper, wonderfully executed and beautifully and logically written, examining the mechanistic question of the function of histone binding to LDs during *Drosophila* development. Using a variety of fluorescent microscopy techniques, the authors show data supporting the model that binding of H2Av to LDs functions to slow nuclear import and prevent toxic effects of too much histones.

With respect to novelty and mechanism, the paper falls short. For novelty, the concepts are not particularly new as the authors have already shown data in a previous report that LDs serve to buffer H2Av. What is new is a systematic dissection of the model using various fluorescent microscopy techniques to investigate the different populations of H2Av, and this is well done. What is lacking mechanistically is quantitative data comparing the rates of binding to LDs with nuclear import, etc. Given that the paper is not entirely novel, more conclusive mechanistic data, i.e. kinetic data that support the model, is at minimal required.

Essential revisions:

Despite the lack of novelty, the reviewers agreed that the paper could be acceptable if a quantitative model was proposed and rigorously tested.

Specifically, what is needed is an model that is explicit (mathematical), integrating biochemical measurements of abundances (of H2Av, other LD bound histones, *Jabba* and import machinery), as well as binding constants (e.g. between H2Av and *Jabba*).

If the authors think that they can substantially "nail" the testing of this model quantitatively, the reviewers agreed the paper could in principle be acceptable.

---

## [Author Response]

With respect to novelty and mechanism, the paper falls short. For novelty, the concepts are not particularly new as the authors have already shown data in a previous report that LDs serve to buffer H2Av. What is new is a systematic dissection of the model using various fluorescent microscopy techniques to investigate the different populations of H2Av, and this is well done.

Thank you. We appreciate both the recognition of this novel aspect and the vote of confidence in the execution. Although the concept of LD-mediated histone buffering had previously been proposed, the mechanistic basis for buffering was unknown. As we point out in the text, the previously most appealing model of buffering (that the organism somehow monitors histone production and demand and releases/captures histones as needed for an appropriate supply) was disproven by our results. Here we demonstrate that buffering is based on the dynamic sequestration of histones and show that it is developmentally regulated. In addition, we discovered that H2Av is under minimal, simplified regulation that fails to include feedback control, starkly contrasting histone regulation in other systems i.e., we had previously hypothesized that LDs represent a regulator of H2Av, but now provide evidence that they are, essentially, the H2Av regulator in early embryos.

What is lacking mechanistically is quantitative data comparing the rates of binding to LDs with nuclear import, etc. Given that the paper is not entirely novel, more conclusive mechanistic data, i.e. kinetic data that support the model, is at minimal required.

In response to this feedback, we have now developed a kinetic model for histone buffering and test it quantitatively.

Essential revisions:Despite the lack of novelty, the reviewers agreed that the paper could be acceptable if a quantitative model was proposed and rigorously tested.Specifically, what is needed is an model that is explicit (mathematical), integrating biochemical measurements of abundances (of H2Av, other LD bound histones, Jabba and import machinery), as well as binding constants (e.g. between H2Av and Jabba).If the authors think that they can substantially "nail" the testing of this model quantitatively, the reviewers agreed the paper could in principle be acceptable.

We have developed a kinetic model (presented in Figure 8, Figure 8—figure supplement 1, Appendix 1, and detailed in the subsection “Devising a kinetic model for LD-mediated H2Av buffering”) to explain the buffering effects of lipid droplets on nuclear accumulation of H2Av. This model incorporates concentrations of the different H2Av pools (i.e., free, LD-bound, DNA-bound), in addition to concentrations of H2Av bindings sites on LDs (i.e., *Jabba*) and H2Av binding sites on DNA. Furthermore, we define rate constants for both the association and dissociation of H2Av with LDs and DNA. We deliberately do not include other molecular players because our goal is to understand to what extent a minimalist, stripped-down model is sufficient to explain H2Av nuclear accumulation in early *Drosophila* embryos, i.e. we are explicitly attempting to model a situation in which the nuclear import machinery and other histones are not limiting or regulated (see subsection “Does H2Av exchange between LDs mediate the buffering role of LDs?”, last paragraph and Figure 11A).

By incorporating a number of experimental observations, we were able to solve the differential rate equations described in Appendix 1. Our solution (Equation 1 in main text, Equation 6 in Appendix 1) relates the rate of formation of DNA-bound H2Av with total H2Av levels, as well as concentrations of H2Av bindings sites on LDs and DNA. We then tested key predictions of this model (predictions presented in Figure 8—figure supplement 1). To really “nail” the testing of this model quantitatively, we now include nuclear import rates of H2Av with varying *Jabba* dosage (including with *Jabba* over-expression), using FRAP, and find the relative rates of nuclear import of H2Av to match those predicted by our model for 1x, 2x, and 3x *Jabba* dosage (subsection “Nuclear H2Av levels depend on LD buffering capacity”, second paragraph, Figure 9E, F). For 0x *Jabba*, we find an intriguing discrepancy that provides additional insights into this experimental system (see the last paragraph of the aforementioned subsection). The full derivation of our model is described in Appendix 1.